# Parameter sensitivity analysis of a 1-D cold region lake model for land-surface schemes

José-Luis Guerrero[1,2], Patricia Pernica[1], Howard Wheater[1], Murray Mackay[3], and Chris Spence[3]

[1]Global Institute for Water Security, National Hydrology Research Centre, 11 Innovation Boulevard, Saskatoon, SK, Canada.
[2]Norwegian Institute for Water Research, Gaustadalléen 21, 0349 Oslo, Norway.
[3]Science and Technology Branch, Environment and Climate Change Canada, 4905 Dufferin Str., Toronto ON, M3H5T4, Canada.

*Correspondence to:* José-Luis Guerrero (jlg@niva.no)

**Abstract.** Lakes might be sentinels of climate change, but the uncertainty in their main feedback to the atmosphere —heat-exchange fluxes— is often not considered within climate models. Additionally, these fluxes are seldom measured, hindering critical evaluation of model output. Analysis of the Canadian Small Lake Model (CSLM), a one-dimensional integral lake model, was performed to assess its ability to reproduce diurnal and seasonal variations in heat fluxes and the sensitivity of simulated fluxes to changes in model parameters, i.e. turbulent transport parameters and the light extinction coefficient ($K_d$). A C++ open-source software package, Problem Solving environment for Uncertainty Analysis and Design Exploration (PSUADE), was used to perform Sensitivity Analysis (SA) and identify the parameters that dominate model behavior. The Generalized Likelihood Uncertainty Estimation (GLUE) was applied to quantify the fluxes' uncertainty, comparing daily-averaged eddy covariance observations to the output of CSLM. Seven qualitative and two quantitative SA methods were tested and the posterior likelihoods of the modeled parameters, obtained from the GLUE analysis, were used to determine the dominant parameters and the uncertainty in the modeled fluxes. Despite the ubiquity of the equifinality issue —different parameter-values combinations yielding equivalent results— the answer to the question was unequivocal: $K_d$, a measure of how much light penetrates the lake, dominates sensible and latent heat fluxes and the uncertainty their estimates is strongly related to the accuracy with which $K_d$ is determined. This is important since accurate and continuous measurements of $K_d$ could reduce modeling uncertainty.

## 1 Introduction

While lakes only cover around four percent of the Earth's land surface (Verpoorter et al., 2014; Cael and Seekell, 2016), their impact on the climate system is disproportionate to their coverage (Williamson et al., 2009). Lakes exert their influence at different time scales. In the long term, down to the seasonal scale, they interact with the climate system through, e.g., their influence on the global carbon balance (MacKay et al., 2009; Tranvik et al., 2009). They also provide more immediate feedback through mass and energy exchanges with the atmosphere. There exists an array of processes, some of them interacting, that modulate the impact of lakes, working at different time-scales (Pérez-Fuentetaja et al., 1999; Kalff, Jacob and Downing, 2002; Tanentzap et al., 2008).

Models of such real-world systems are necessarily simplified conceptualizations, making them tractable (Beven and Germann, 2013). Inferences drawn from models will be plagued by issues such as epistemic gaps, data uncertainties, or even computational artifacts (Clark and Kavetski, 2010; Westerberg et al., 2011; Beven and Westerberg, 2011). Taking modeling uncertainties into account should be of fundamental importance, but they might not be considered due to lack of evaluation data, computational limitations or a worrying avoidance of the issue, among other things.

In order to represent the influence of lakes in the climate system, lake models are embedded into land-surface schemes that can in turn be coupled to regional or global climate models. These cascading systems are linked through their inputs and outputs, sometimes considering feedbacks (Shrestha et al., 2014). The uncertainties in the couplings, even when propagated through simple systems, can produce a wide range of potential outputs (Beven and Lamb, 2014). These, and other uncertainties, make mimicking the hydrological system in coupled land-surface/climate models a practical challenge for current modeling systems (Kundzewicz and Stakhiv, 2010). A first step in improving these systems would be to quantify the uncertainty of the linkages, and if possible reduce it.

In general, hydrological aspects of the climate system were effectively ignored in early modeling efforts (Phillips, 1956) or summarily represented (Manabe, 1969). This was mostly due to computational limitations but also to limited process understanding (Koster and Suarez, 1992; Koster et al., 2000). Up until the 1990's, most open water surfaces were not resolved in climate models (Pitman, 1991): only large lakes could be represented (Bates et al., 1993). These early conceptualizations are simplistic, viewing lakes as saturated soils with modified roughness and albedo (Pitman, 1991) or as slabs of water with no differentiated mixing (Ljungemyr et al., 1996), and ignore the internal thermal structure of lakes, which influences fluxes to the atmosphere (MacKay, 2012).

Lakes are not inert masses but living systems. From the point of view of atmospheric feedbacks, ecosystem function is more than just ontologically relevant and is a controlling factor for heat exchange. Previous studies illustrate the feedback between phytoplankton and thermal structure, via light extinction modulation (Tilzer, 1983, 1988; Mazumder et al., 1990; Rinke et al., 2010). Thermal stratification modulates oxygen concentrations and therefore ecosystem function (Elçi, 2008). Palaelogical studies of lake ecosystems show they are highly sensitive to environmental change (Eggermont and Martens, 2011). Understanding energy feedbacks between lakes and the atmosphere, or at least estimating the associated uncertainties, is of central importance to diagnose the potential impacts of change. Accounting for these uncertainties could anchor the results of studies such as Samuelsson et al. (2010) and Rouse et al. (2005) who show how lakes impact regional climate and contribute to greenhouse gas emissions (Stepanenko et al., 2011; Tan et al., 2015).

More than half the global lake area consists of small lakes (Downing et al., 2006), which might not be resolved at the typical scales of global or mesoscale models. Furthermore, the spatial pattern of mass and energy fluxes directly influence the evolution of the atmospheric boundary layer, thus compounding the issue (Shrestha et al., 2014): There is a distinct difference both in the timing and the magnitude of fluxes between open water surfaces and the atmosphere compared to land (Halldin et al., 1999). The magnitude of these fluxes can be a function of several factors, such as lake area (Woolway et al., 2016) and the latitude of the lake (Woolway et al., 2017). The clarity of the lake seems to be the dominant factor (Heiskanen et al., 2015; Woolway et al., 2016; Rose et al., 2016).

With different albedo, heat capacity, and surface roughness compared to the surrounding land areas, lakes also provide more immediate feedback through transfer of heat and moisture exchanges with the atmosphere (e.g., MacKay et al., 2009; Xiao et al., 2013; McGloin et al., 2014b). While some studies have performed direct measurements of latent and sensible turbulent heat fluxes from eddy covariance systems over lakes and reservoirs (e.g., Blanken et al., 2000; Vesala et al., 2006; Blanken et al., 2011; Nordbo et al., 2011; McGloin et al., 2014a) these measurements can be difficult and expensive and as such improved modelling approaches are necessary (e.g., McGloin et al., 2014b).

The Canadian Small Lake Model (CSLM; MacKay, 2012), a 1D, deterministic, bulk-mixed layer model, was developed to integrate within the Canadian Land Surface Scheme (CLASS; Verseghy et al., 1993; Verseghy, 2007), which can in turn be coupled to regional climate models as well as large scale hydrological models. CLASS resolves heterogeneity in the landscape using a mosaic approach (Koster and Suarez, 1992) where the CSLM acts as a tile generating its own flux exchange with the atmosphere. Previous work with CSLM has demonstrated its ability to reproduce surface temperatures over a range of conditions within different lakes of the Experimental Lake Area (MacKay, 2012). Evaluation of the model in terms of surface heat fluxes is generally lacking, however. In this paper we use observed micrometeorological flux data from a small lake (Landing Lake, 62.5°W, 114.4°N, Surface Area = 1.12 km$^2$) in the North West Territory of Canada (Fig. 1) to explore model performance, and demonstrate the capacity of alternative methods of sensitivity analysis to identify the relative significance of model parameters and their impact on uncertainty in the simulation of lake-atmosphere energy exchange.

Below we briefly introduce the case-study application and the model, and then present and discuss alternative methods for model sensitivity analysis, drawn from the PSUADE toolbox. The paper presents a comparative analysis of these alternative approaches, and concludes with a summary of key findings and general discussion of the implications.

## 2 The Canadian Small Lake Model

CSLM is a 1-D (with depth as the vertical coordinate axis), bulk-mixed layer model that outputs the temperature profile within the water column and sensible and latent heat at each time step. The surface boundary is set using atmospheric conditions while the boundary at the base of the lake is adiabatic. The model is forced at each time step with meteorological data. Using an initial temperature profile, the surface energy balance is solved at the boundary while the conductive and radiative heat flux is solved at each depth intervals. From this heat flux and using the 1-D heat equation, the temperature profile of the lake is recalculated for the current time step. At this stage if there are any static instabilities in the temperature profile, mixing occurs using an integrated turbulent kinetic energy (TKE) approach. This generates a final temperature profile. The surface temperature from the profile is then used within the bulk aerodynamic formulas to calculate both sensible and latent heat fluxes. A complete description of the model can be found in MacKay (2012).

Along with the initial temperature profile and standard meteorological forcing, the light extinction coefficient $\left(K_d,\ \mathrm{m}^{-1}\right)$ for a given lake is also required. The light extinction coefficient, $K_d$ (Table 1), is a measure of how light in the visible spectrum attenuates through the water column; a measure of the transparency of the lake. Low values of $K_d$ indicate a clearer lake where light can penetrate deep into the water column. Higher values of $K_d$ indicate a more turbid lake where light attenuates much

closer to the surface. The value of this parameter has direct implications for the temperature profile within a lake and thus both the sensible and latent heat fluxes. The mechanisms underlying the thermal structure are complex. From a purely mechanistic perspective water clarity affects lake hydrodynamics (Persson and Jones, 2008), even under a 24-hour period (Woolway et al., 2015). The thermal structure further depens depends on lake morphometry (Woolway et al., 2016) and is compounded by biogeochemical processes, such as browning waters (Roulet and Moore, 2006) and ecosystem function (Tilzer, 1983, 1988; Mazumder et al., 1990; Rinke et al., 2010). Within CSLM, shortwave extinction is exponential with depth following Beer's law, with $K_d$ identified as the e-folding depth.

The turbulence subroutine in CSLM is used to determine the mixed layer depth; the depth over which active mixing occurs homogenizing the temperature profile. This is the final step within the model before the temperature profile and fluxes are output. The change in mixed layer depth is calculated by assessing the change in TKE. This change is determined through the competition between energy input terms which act to increase the mixed layer depth and loss terms which act to decrease the depth of the mixed layer. Energy input terms include wind driven stirring and buoyancy fluxes ($F_q$), transport of TKE to the thermocline ($F_i$) which acts to erode it, and shear production at the base of the mixed layer ($F_s$). Loss terms include energy dissipation ($F_d$), entrainment of deeper water at the thermocline ($F_p$) and sinks of TKE within the mixed layer ($F_L$). In this scheme the modeled integrated TKE budget is used to determine the turbulence within a mixed layer of uniform properties and depth $h$ (Imberger, 1985; Spigel et al., 1986). This is expressed as:

$$\frac{d}{dt}\left(\frac{1}{2}hE_s\right) = \frac{h}{2}\frac{dE_s}{dt} + \frac{E_s}{2}\frac{dh}{dt} \tag{1}$$

where $\frac{E_s}{2}$ is the average TKE per unit mass. To solve the energy budget, the terms on the righ-hand-side are re-written as the sum of relevant turbulent processes:

$$\frac{h}{2}\frac{dE_s}{dt} = F_q - F_d - Fi \tag{2}$$

$$\frac{E_s}{2}\frac{dh}{dt} = F_i + F_s - Fp - F_L \tag{3}$$

This parameterization involves five empirical turbulent coefficients (Table 1). With the exception of the entrainment term, $F_p$, the definition of each of the terms contains a constant empirical coefficient; $c_n$ for surface mechanical input, $c_e$ for TKE dissipation, $c_f$ for TKE transport to the thermocline, $c_s$ for shear production and $c_L$ for the sinks of TKE. Experiments detailed in Spigel et al. (1986) yield a range of values for $c_n$, $c_f$, $c_e$ and $c_s$. A set of consistent values was chosen by Rayner (1980) and have been used subsequently by Spigel et al. (1986) and MacKay (2012); $c_n = 1.33, c_s = 0.2, c_e = 1.15, c_f = 0.25$. As mentioned above, MacKay (2012) chose $c_L = 0.235$ based on experimental data.

Both turbulent mixing and $K_d$ directly control the output of temperature and heat fluxes. In addition, there is also a considerable amount of uncertainty in both. In the case of the turbulent subroutine, the values of the empirical constants while

consistent have never been investigated. The value of $K_d$ on the other hand carries uncertainty due to limitations in its measurement both spatially and temporally. Although measurement technology has long been available (Poole and Atkins, 1929), continuous measurements of $K_d$ are, to the best of our knowledge, relatively scarce despite the relative affordability of cosine collectors, perhaps currently the most widespread technology; see Frankovich et al. (2017) for a recent application. Also, point
measurements can be taken through more or less direct proxies, such as the Secchi disk depth (Tyler, 1968) and Dissolved Organic Compounds (DOC) concentrations (Ask et al., 2009). The technology is in fact evolving (Chudyk and Flynn, 2015). Alas, despite the availability of measurement tools, $K_d$ might sometimes be an afterthought, as in the case here presented, where it was determined from DOC concentrations.

## 2.1  The Lake

Data from Landing Lake in Baker creek (NWT, Canada; Fig. 1) were used to evaluate temperature profiles and heat fluxes produced by CSLM. Landing Lake is a small freshwater lake with a surface area of 1.12 km$^2$. While no comprehensive bathymetry measurements have been taken on Landing Lake, depths in the main body of the lake during installations of thermistors and pressure transducers over the course of this study and others are consistently 4 m. The lake's two southern arms are shallower, near 1.5 m, as can be seen by the change in colouration in Fig. 1. Concentrations of DOC are high,
resulting in an expected $K_d$ value of $\sim 2\text{m}^{-1}$ (Spence et al., 2003). Due to the shallow depth and high value of $K_d$, the lake does not form a seasonal thermocline, but diurnal thermoclines were observed in the temperature data.

Meteorological, radiation and turbulent flux measurements were taken from a climate station installed on a bedrock outcrop island that was first described in Granger and Hedstrom (2011). This location provided fetch distances that ranged from 150 m to 900 m. Data from the station were obtained for the open water periods 2007–2009. Turbulent fluxes of sensible and latent heat
( W·m$^{-2}$, positive upward from the surface) were calculated from 10 Hz measurements of the vertical wind speed (m·s$^{-1}$), air temperature (°C), and water vapour density (g·m$^{-3}$). Wind speed was measured using a 3-D ultrasonic anemometer (Campbell Scientific CSAT-3), while water vapor density was measured using a krypton hygrometer (Campbell Scientific KH20) located 25 cm away and at the same height as the sonic anemometer. The statistics (means and covariances) of the high-frequency data were collected and processed at 30-min intervals using a datalogger (Campbell Scientific CR3000). Corrections to the eddy
covariance measurements include 2-D coordinate rotation (Baldocchi et al., 1988), air density fluctuations (Webb et al., 1980), sonic path length, high-frequency attenuation, and sensor separation (Massman, 2000; Horst, 1997). Associated 30 minute average meteorological observations included horizontal wind speed (m·s$^{-1}$) measured with a Met-One 14A cup anemometer, air temperature (°C) and relative humidity (%) measured with a Vaisala HMP45C thermohygrometer. Incoming and outgoing short wave radiation (W·m$^{-2}$) were measured with paired upward and downward facing Li-Cor LI200S pyranometers. A Kipp
and Zonen NRLite was mounted 1.04 m above the water to measure net radiation (W·m$^{-2}$). Because of the homogenous nature of the lake bathymetry one vertical array of Onset pendant thermistors was deployed 300 m northeast of the island measuring half hourly water temperature at 4 depths (0 m, 0.5 m, 1 m, 2 m) in 2007 and 2008 and at 3 depths (0 m, 0.5 m, 1.5 m) in 2009.

## 3  Sensitivity Analysis — an overview

When considering model-performance analysis, particularly in the case of complex models, it is important to note that one or more model parameters might exert more or less influence on one or more model outputs. Some of these parameters may be observable and/or measured while others may have dubious physical interpretation. How to specify model parameters is not a trivial issue (Gupta and Sorooshian, 1985; Stefanski, 1985; Wagener et al., 2003). Over-parameterization, parameter interactions, erroneous evaluation data, and computational errors, are all causes of equifinality (Beven, 2006): different parameter-value combinations yielding nigh-indistinguishable results. Sensitivity Analysis (SA) provides a way to mitigate the equifinality issue by identifying the parameters that dominate model performance. Unimportant parameters may be used to reduce dimensionality, paliating equifinality with minimal impact on performance (Huang and Liang, 2006; van Werkhoven et al., 2008, 2009). Better constraints on important parameters, e.g. through improved measurements, may result in uncertainty reduction.

There are many different approaches for SA, see Gan et al. (2014) and Song et al. (2015) for a thorough discussion. Song et al. (2015) in particular provides an exhaustive overview of the state-of-the-art. In general terms, SA methods can be classified as global and local. Local measures assess model response by varying one parameter at a time while global measures vary all parameters simultaneously. Local measures do not account for possible parameter interactions, but are computationally lighter since, all other conditions being equal, they require fewer evaluations. The less demanding method is in fact differential SA, which uses partial derivatives or finite differences at a location —parameter-value combination— of interest.

If the location of interest within the parameter space is not known a-priori, a common occurrence given the equifinality issue, then random sampling global SA measures are preferred. However this necessitates more model evaluations since instead of varying just one parameter or looking at a specific location they are based on exploring the entirety of the feasible parameter space. The Generalized Likelihood Uncertainty Estimation (GLUE; Beven and Binley, 2014) is a global SA method that evolved from the work of Hornberger and Spear (1981) and consists of randomly sampling the prior parameter space and evaluating the performance of the model at each random parameter-value vector, selecting behavioral (well-performing) vectors after either subjective thresholding (Li et al., 2010), or using measurement error as a splitting criteria: the limits of acceptability approach (Coxon et al., 2014).

The projection of the multidimensional parameter-value vector into a plane defined by one of said parameters and the corresponding performance ("dotty plots") can then be used to define a one-dimensional frequency distribution that is indicative of the parametric sensitivity. Furthermore an estimate of the uncertainty of model simulations can be obtained by weighing them according to the performance, deriving uncertainty bounds. The GLUE approach, however is computationally inefficient, especially since strong information about prior parameter distributions is often unavailable and random uniform sampling required.

Computational performance of global SA can be improved using the Design Of Experiment approach (Tong and Graziani, 2008), which consists of two steps. First, while still random, sampling is designed to efficiently cover the prior parameter space, or is tailored specifically for a given global SA method. Second, variation in model performance is attributed to the variation

of different parameters, see Gan et al. (2014). The relative ranking of parameters and quantitative attribution of model-output variance might change depending on the sampling technique and global SA method (Gan et al., 2014).

Tong (2013) developed the PSUADE (Problem Solving environment for Uncertainty Analysis and Design Exploration) package that provides a collection of tools to perform uncertainty quantification and sensitivity analysis. PSUADE has been used to produce technical reports related to the modeling of explosives (Hsieh, 2006; Wemhoff and Hsieh, 2007), to the modeling of a two-dimensional soil-foundation structure-interaction (Tong and Graziani, 2008), and to the modeling of an electrostatic micro-electromechanical-system switch (Snow and Bajaj, 2010). Tong and Graziani (2008) use PSUADE to produce a book chapter exploring uncertainty quantification for multi-physics applications.

All the listed applications only focus on a subset of the methods available in PSUADE. Similarly, in hydrological applications, most SA studies focus on a single method, see Table 3 in Song et al. (2015) for an overview of recent applications, sometimes disregarding global SA in favor of local SA: e.g. most applications of PEST (Skahill and Doherty, 2006) where the number of calibration parameters can be reduced using local SA.

The present study was based on a combination of three factors underlining its relevance. Firstly, by building upon existing literature that stresses the importance of lake clarity in modeling heat transfers (Heiskanen et al., 2015; Rose et al., 2016; Woolway et al., 2016), evaluating against measured fluxes, as done by Deacu et al. (2012) for large lakes.

Secondly, while the difficulty in finding adequate parameterizations for land-surface schemes has been recognized (Hogue et al., 2005; Duan et al., 2006; Demarty et al., 2005), and the importance of incorporating observational data has been underlined (Liu et al., 2005), little effective attention has been placed on uncertainty analysis in this kind of physical modeling. Regarding lakes, inroads have been done, but with respect to water quality (Missaghi et al., 2013).

Thirdly, PSUADE is a recently available tool that provides the mechanisms to perform the kind of exhaustive SA pioneered by Gan et al. (2014) that allows easy testing of different methods within a single package and hence provides more robust results. Furthermore quantifying the uncertainty in the connecting fluxes of the different components of a modular system (in this case a land-surface scheme) should be one of the first steps, often not performed, in an overall uncertainty assessment. This paper also represents a start in that direction. Our concrete objectives were to find:

a) What were the parameters that dominated model performance, in terms of latent and sensible heat fluxes, evaluated with two different objective functions, the Nash-Sutcliffe Efficiency (NSE; Nash and Sutcliffe, 1970) and the Mean Absolute Error (MAE)?

b) What was the uncertainty in the modeled fluxes, which was quantified using the GLUE methodology?

## 3.1 SA methods

The purpose of this section is to describe without going into mathematical detail the different methods used for sensitivity analysis, emphasizing the assumptions each one makes. A formal description of the different methods can be found in Gan et al. (2014).

It should be kept in mind that there are different ways of categorizing SA methods (Song et al., 2015) and that there is no consensus regarding terminology (Razavi and Gupta, 2015). The methods used in this paper were all global —the combined effect of multiple parameters was considered— and the term 'sensitivity' itself was meant as a ranking of the impact that model parameters had on model performance, obtained from comparison of observed and modeled data. In broad terms, the global SA methods applied are classified as qualitative methods that provide a relative ranking of parameter sensitivity and quantitative methods that attempt to explain how much of the variance in the model performance is explained by the variance in each individual parameter or combination of parameters.

### 3.1.1 Description

Besides their ability to screen the most important parameters, the common thread between qualitative methods is that they require relatively fewer model runs, compared to quantitative ones. They might however differ in their conceptual approach. For instance the Spearman rank correlation (SPEAR; Spearman, 1904) and the Standard regression coefficient (SRC; Galton, 1886) share a conceptual framework: they are regression methods, that simulate performance as a linear combination of parameter values.

SPEAR bases its sensitivity rankings on the degree of linear correlation between each individual parameter and performance. SRC stipulates a predictive model as a linear combination of all parameter values. The SRC value for each parameter is obtained by normalizing the coefficients of the predictive model. It should be noted that the predictive model needs not be linear, but often is, as was the case here. A downside of the regression methods is that their robustness is dependent on their predictive capability (Yang, 2011).

Another conceptual approach is to view the partial derivatives of model performance with respect to model parameters as indicators of parameter sensitivity: the steeper the response surface around a given point, the more sensitive the parameter in that region. An analytical solution would allow explicit evaluation over the entire parameter space but that is a practical impossibility for most, if not all, models. Instead numerical approximations are computed at selected points and averaged to give an indication of the relative sensitivity of the model parameters. This is the Morris one-at-a-time (MOAT; Morris, 1991) approach. The sensitivity is evaluated by computing both the mean (MOAT-1) and the standard deviation (MOAT-2) of the partial derivatives at selected sample points. It is a robust method in the sense that no assumptions are made for the relationship between model parameter values and performance.

The final conceptual approach consists in assuming a functional relationship between performance and parameters. The sensitivity is assessed by evaluating whether or not the inclusion of a given parameter in the functional relationship affects the performance simulation. These methods fall under the denominational umbrella of response surface modeling (RSM). Examples of such methods are: Multivariate Adaptive Regression Splines (MARS; Friedman, 1991), Delta-Test (DT; Pi and Peterson, 1994), Sum-of-Trees (SOT; Breiman et al., 1984; Chipman et al., 2012), and Gaussian Process (GP; Gibbs and MacKay, 1997). They are not robust in the sense that they rely on a-priori assumptions about the nature of the response surface, and will produce reliable results only if those assumptions are met.

MARS is an extension of the concept of linear models to a multidimensional setting and consists of fitting, through linear regression, (hyper-)planes to the response surface of the model. It is in essence an extension of the recursive partitioning approach to regression: determining the breaks for the piecewise linear fits from the data. The relative importance of the parameters is determined by dropping them in turn from the regression and reevaluating the performance: the bigger the drop in performance, the more important the parameter.

DT was originally developed for time series modeling, the basic premise being that a chaotic dynamic system can be reconstructed from sequences of observations of its state (Pi and Peterson, 1994). Eirola et al. (2008) is an example of the DT method parameter-screening tool: the subset of parameters that minimize the variance in the noise (difference between observed and modeled performance) are seen as the most sensitive ones. Testing all possible parameter subsets is computationally infeasible and the PSUADE package chooses the best 50 subsets for scoring. Furthermore, the method itself is computationally demanding since it might require operations on large matrices and can be affected by numerical instabilities.

The premise of SOT is that model parameters can be used to simulate performance based on a binary decision tree: each parameter can be used to partition the parameter space into two areas with different responses and the sum of all different partitions used to predict performance. The number and partition setup is determined through a recursive binary division of the parameter space. The required number and ordering of the partitions is evaluated by comparing the residuals between the tree-predicted performance and the computed one, until a convergence criteria is met. The relative importance of each parameter is proportional to the number of splits in the tree that include that parameter.

GP assumes performance follows a multivariate normal distribution, characterized by the means and the covariance matrix of the different parameters. Whereas the means of the parameters are dependent on their relative scalings, the normalized covariances are not and this allows to compare the degree of change in the response along the different dimensions. The relative degrees of change along the different dimensions are an indicator of parameter sensitivity.

These qualitative measures do not explain how much of the performance variance is due to a given parameter (or parameter-interaction). Quantitative measures such as Fourier Amplitude test (FAST; Cukier et al., 1973), McKay main (McKay-1) and two-way (McKay-2) interaction analysis (McKay et al., 1999) and Sobol sensitivity indices (Sobol, 1990, 2001) provide such quantitative assessment. All quantitative methods are variance-based methods that use an ANOVA-like decomposition (Fisher, 1925) to identify the subset of parameters that dominate performance simulation, but differ on the way performance is simulated.

In FAST, model performance is expressed as a Fourier series, incorporating different model parameters. Since the model is a function of several parameters, a multi-dimensional integral is required to evaluate the Fourier coefficients. This is solved by making it one-dimensional through application of the ergodic theorem. Of the methods listed here, it is the one that requires the least amount of model runs.

The McKay method attempts to find the subset of parameters that better approximates the performance variance by computing the ratio between the performance variance for a subset of parameters and the performance variance for all parameters, which is a measure of the relative importance of each subset. The performance variance is computed through random sampling of the parameter space, making the method non-parametric The subset that maximizes the ratio is considered the most sensitive

subset. McKay et al. (1999) extended the concept to account for two-way parameter-interactions (McKay-2) and their effect on the performance variance, assuming the parameters to be uncorrelated.

In the Sobol method, model performance is decomposed into summands of functions of the parameters, in increasing order of dimensionality. Assuming ortogonality of all summands permits expressing the performance in terms of a sum of conditional expected values. Further assuming that said sum is square integrable makes it possible to equate performance variance to a sum of variances and covariances of model parameters. The effect of model parameters on model output can then be decomposed into first-order indexes, where the contribution of the variance of each individual parameter on model output be quantified and can also include the interactions of model parameters (the covariances). The number of interactions to include range from second order, where just two-way parameter interactions are included, to total-effect, where all possible interactions are accounted for. The final result is a ratio that shows how much of the output variance can be explained by a parameter or combination of parameters.

## 3.2 Uncertainty analysis with GLUE

GLUE (Beven and Binley, 2014) is a widely applied method, particularly in hydrology, used to evaluate parametric sensitivity and to quantify parametric uncertainty. This paper focuses on the latter aspect in order to provide an estimate of the impact of uncertainty on the modeled fluxes.

GLUE starts with a random sampling of the parameter space and subsequent computation of the simulation-performance for each random parameter. The random runs are then classified into behavioral (well-performing) or non-behavioral according to either subjective (e.g. threshold value) or objective criteria (limits-of-acceptability; Coxon et al., 2014). The behavioral simulations are then weighted according to the performance and uncertainty bounds extracted from weighted simulations: e.g. at each time step the 0.05 and 0.95 percentiles of the likelihood weighted simulations can be extracted and considered to be the 95% confidence interval.

To each combination of parameter values corresponds a performance and the projection of the (multidimensional) parameter-values against performance along one dimension, or parameter-axis, produces what are commonly known as "dotty plots" which can give an idea of parametric sensitivity (Beven, 2006).

## 4 Methods and performance metrics

The CSLM simulates heat-transfer fluxes —latent and sensible heat— at the boundary between lake surface and the atmosphere. The model was run with half-hourly forcings and the resulting simulations were temporally aggregated to evaluate against daily flux data, which was obtained through integration of hourly eddy-covariance measurements. The aggregation was necessary because of inherent limitations of the higher-frequency daily covariance data, that were available for the period 12/06/2007–18/10/2007. Two performance metrics were used for the evaluation, MAE and NSE:

$$\text{MAE} = \frac{\sum_{i=1}^{n} |O_i - S_i|}{n} \tag{4}$$

$$\text{NSE} = 1 - \frac{\sum_{i=1}^{n} (O_i - S_i)^2}{\sum_{i=1}^{n} (O_i - \bar{O})^2} \tag{5}$$

where $n$ is the number of time steps at which the model was evaluated, $O_i$ was the observed value at time step $i$, $S_i$ was the simulated value at time step $i$, and $\bar{O}$ is the mean of the observed values. The rationale behind the choice was to contrast the propensity of NSE to prioritize better-fitting high-values (Krause et al., 2005), sometimes to the detriment of other ranges, by using MAE as a contrasting metric, less dependent on high-value fit. The entire period with available data was used for the computation of the performance measures. The metrics were computed after aggregating the hourly data to a daily time step.

MAE and NSE were the basis of all tested SA methods (Table 2). The PSUADE (Tong, 2013) package is a tool that assembles SA methods under a unified computational framework, thus facilitating exhaustive testing in the vein of Gan et al. (2014), who tested the impact of sampling, in terms of frequency and technique, and different SA methods in the identification of sensitive parameters for a hydrological model. The general procedure for PSUADE is: a) generate random samples, b) run the model and compute the performance for all the samples, and c) compute SA metrics from the obtained performances. The performance used was the average performance for the simulation of latent and sensible heat.

From a pragmatic point of view, the choice of SA method might depend on the computational requirements of the tested model. Different methods might require different number of simulations to achieve consistent results, as shown by Gan et al. (2014). The number of simulations required is primarily a function of the complexity of the response surface, a factor difficult to control, but also depends on the sampling scheme and the theoretical basis of the SA method. Different schemes represent different ways of exploring the parameter space and some are better suited than others depending on the purpose (McKay et al., 1979).

The CSLM was sufficiently fast to run, less than a second for 128 days of half-hourly data, so that the number of simulations required by the different methods was not a limitation. Therefore we ran as many simulations as necessary to obtain consistent results (Table 3). The choice of sampling method was based on those used by Gan et al. (2014) in their experiment.

Seven qualitative and two quantitative SA methods were tested to identify the sensitive parameters in CSLM. The parameters chosen for the test are the ones deemed a-priori to have physical significance for the thermodynamic functioning of the lake. The range of the parameters was based on physically plausible values (Table 1).

Finally, in order to broadly estimate the impact of parametric uncertainty on heat-flux simulation, the GLUE procedure was applied to CSLM for the parameters, and ranges, listed in Table 1. A total of one million simulations were performed and the top 10% selected as behavioral in order to compute the uncertainty bounds. For MAE the lower the value, the better the performance and therefore the inverse of the computed MAE was used as the likelihood when performing GLUE. The NSE values were used unmodified for the weighting.

## 5    Results

### 5.1    Qualitative measures

We first performed SA using qualitative methods (Table 2) and obtained a ranking of the model parameters in terms of sensitivity. The results were presented as a color map where the darker tones indicate larger sensitivity (Figs. 2 and 3). Near unanimity was reached, for both MAE and NSE, in identifying $K_d$, the light attenuation coefficient, as the most sensitive parameter for the simulation of heat fluxes (Table 4). The sole exceptions were the SPEAR and SRC methods. A strong assumption for both these methods is that model response varies linearly with input, which is almost never the case for environmental models (Beven 2006): Response surfaces tend to be very complicated (Duan et al., 2006). Another method with strong prior assumptions about model response is GP, that fits the response surface to a multivariate normal distribution. Global model response is seldom normal due to the overall complexity of the response surface, but if the response is locally linear in a region of the parameter space, then the approximation might be adequate (Kuczera, 1990).

A physical explanation of the results obtained here, that pinpoint $K_d$ as the most sensitive parameter in terms of simulating heat transfers, can be inferred from the fact that the lake is shallow and therefore it is perhaps not surprising that the turbulent transfer coefficients exert no major impact on such processes. A similar analysis —not shown here— studying parameter sensitivity with respect to the thermal structure of the lake also highlighted the importance of the $K_d$ parameter for the thermal balance of the lake.

### 5.2    Quantitative measures

It is evident (Figs. 2 and 3) that model response, in terms of simulating heat fluxes, is dependent on the value of the $K_d$ parameter. None of the qualitative measures, however, do explain how much of the variance in the response can be explained parameter correlations, the combined effects of two or more parameters with respect to output variance.

McKay-2 (Figs. 4 and 5) ranked two-way correlations, which is the sum of first and second order effects of different parameters combinations on performance. The importance of $K_d$ was once again incontestable: no other parameter combinations besides those containing $K_d$ were more important for the simulation of heat transfer. Put in another way, this also meant that $K_d$ by itself was more important than any other parameter combination. This is especially revealing since even parameters with low main-effect may have significant effect on performance through their interaction with other parameters, but here all points to $K_d$ as the main culprit.

In fact, from first order to total-order effects (Figs. 6 and 7), it was $K_d$ that dominated model response both in qualitative and quantitative terms. There was however some difference in the first-order effects estimated with either FAST and SOBOL-1 (Figs. 6 and 7). A conceptual difference between the two is that FAST approximates model output as a Fourier series, which implies a better fit might be obtained in time series with a degree of seasonality whereas SOBOL only relies on expected values computed from a random sample of the data. The downside of the SOBOL analysis is that it requires many more simulations to reach consistent results. There was no reason to expect seasonality for this dataset, as it comprised less than one year. The

model was lightweight enough so that computation time was not a factor for the SOBOL analysis. Therefore the SOBOL results, which show a larger importance for $K_d$, were probably more indicative effective of parameter sensitivity.

## 5.3 GLUE

A major criticism of the GLUE procedure is that it can be subjective in the selection of behavioral parameters. As such, the uncertainty bounds presented here (Fig. 8) should be taken with a pinch of salt, since the selection criteria was set as the 10% best performing parameters from the Monte Carlo simulations. This was subjective but allows for a common criteria for the behavioral threshold for MAE and NSE, which are not directly comparable. Also, given the large number of simulations performed, the results were a robust indicator of the possible output range: the ability of the model to reproduce observed data.

The importance of $K_d$ is once again highlighted in the dotty plots (Fig. 9): it was the only parameter that performance was sensitive to. The value of the turbulent transport parameters had little impact on model performance.

Finally, there is a trade-off in the simulation of latent and sensible heat (Figs. 8 and 10) and even considering the generous behavioral threshold that was set, the uncertainty bounds did not always encompass measurements (Fig. 8).

## 6 Conclusions

Most SA methods pinpointed $K_d$ as the most sensitive parameter in terms of simulating latent and sensible heat fluxes. The exceptions were those which had the strongest assumptions with respect to the nature of model response: SPEAR and SRC assume linearity between input (model parameter) and response (performance measure), which was not the case for CSLM. Somewhat surprisingly GP, which also makes assumptions about the shape of the response surface and considers it Gaussian, also showed $K_d$ to be the most sensitive parameter. This either was a false positive or the response surface might have been locally linear, thus justifying the normality assumption (Williams, 1998). None of the other methods makes such strong assumptions about the nature of the response surface, although localized fitting does occur: e.g. stepwise linear for MARS.

This is perhaps not surprising given the recognized importance of the light extinction coefficient in modulating heat transfers (Heiskanen et al., 2015; Woolway et al., 2016; Rose et al., 2016). Given the complex and intertwined processes that can affect light penetration, such as browning waters (Roulet and Moore, 2006) and ecosystem function (Tilzer, 1983, 1988; Mazumder et al., 1990; Rinke et al., 2010), a single measurement of its value might prove insufficient. It might be necessary to rely on continous measurement in order to improve modeling, either through adherence to a parsimony principle (its value need not be modelled if actually measured) or stemming from the need of evaluating the complex processes influencing it value.

The light extinction coefficient is the only parameter considered in this analysis that is directly measurable (Table 1). Being able to measure the most sensitive parameter is a definite advantage: it allows to confidently reduce the number of parameters needing calibration. Furthermore, since $K_d$ is so predominant in terms of model performance that making it a measured instead of a calibrated quantity should facilitate further model evaluation: Whatever variability in the performance that was not a function of $K_d$ should become easier to quantify/analyze since it would not be obfuscated by $K_d$'s predominance.

With respect to large scale applications, such as climate modeling and land-surface schemes, measuring $K_d$ for every lake might be a practical impossibility, but its importance should be stressed and further research devoted to assess its temporal and spatial varibility. Remote sensing might provide a solution to lack of in situ measurements, and is in fact used to provide estimates of the Secchi depth (Torbick et al., 2013), a proxy for the light extinction coefficient.

The model did not perform equally well at different time periods (Fig. 8) and $K_d$ is known to vary over time (Rinke et al., 2010). A first step into assessing causality between these two factors would be to continually measure $K_d$, at least at a daily time step. Such measurements have proved useful in evaluating turbulent transfers over large lakes (Deacu et al., 2012).

There is often a disconnect between experimentalists and a computer modelers (Seibert and McDonnell, 2002, 2013). The conceptual framework behind the present study is a testament to an improvement of those dialogues: the undertaken modeling
approach was based on data from an observation station that was explicitly established to support testing of hydrometorological models over lakes. Such new data facilitated the analysis performed here and it must be stressed that measuring $K_d$ was not part of the objectives of the field campaign. The modeling exercise performed here underlined its importance for the simulation of heat fluxes and represents an argument in favor of its monitoring.

The limitations of SA methods are tightly related to the curse of dimensionality (Saltelli and Annoni, 2010). All SA methods
obviate the issue by randomly sampling the parameter space, implicitly assuming results to be representative of the entire space. This is especially problematic in high-dimensional spaces, where even the simplest of models might face prohibitive computational costs and even the most sophisticated sampling schemes are limited in their coverage of the parameter space.

The uncertainty bounds obtained from the GLUE analysis are nothing but subjective since the behavioral threshold was arbitrarily set the top 10% simulations. Even with this rather lenient criteria, the bounds did not encompass all observations,
especially for sensible heat (Fig. 8). This might be due to the tradeoffs between latent and sensible heat simulation (Fig. 10) and the chosen performance measures, but might also stem from inadequate process representation. The clear tradeoffs in performance for latent and sensible heat might be influenced by the fact while the CSLM surface energy balance is a strongly nonlinear function of the surface skin temperature, both the sensible and latent heat fluxes are linear terms in this relationship. All other things being equal, this leads to a direct tradeoff between them: the capacity of the model to simulate one of the terms
is inversely proportional to its ability to simulate the other, see Fig. 10.

PSUADE is a powerful package that facilitates SA by providing a wealth of approaches, and presents an opportunity that evaluate their appropriateness, a factor of special importance since they are based on different conceptual precepts. Overall the results might depend on the quality of the sampling, but that was not a factor here since the model was lightweight enough for the number of runs not to be a limiting factor. This will definitely not be the case with more complex models. If runtime is an
issue methods like GLUE become nonviable. FAST on the other hand requires relatively fewer simulations to reach consistent results, but makes assumptions about the nature of the response surface. The later is true for most RSM methods: the robustness of the results depend on the validity of these assumptions.

As such, non-parametric methods like McKay or MOAT might be preferable, the cost being large computational require-ments. In any case the shakiest conceptual basis is for the methods assuming linearity in the model response, i.e. SPEAR and
SRC, as that is seldom the case for environmental models. The method that provides the most information about parameter

interaction is the SOBOL method, the cost being numerical instabilities, especially when a large number of parameters is involved. All methods might fail in the presence of singularities or discontinuities in the response surface and if they give the right answer it might be for the wrong reasons.

In all, to recommend a preferred method is difficult and very much depends on the nature of the problem. If the popularity of their use is an indicator, then MOAT and the RSM are the most used in the literature (Song et al., 2015). It is the authors' opinion that the SOBOL method provides the most complete insight, but is difficult to apply in high-dimensional spaces and is prone to numerical instabilities.

Despite the shortcomings, the answer to our original question was clear: $K_d$ is undeniably the most sensitive parameter in the simulation of heat fluxes.

*Acknowledgements.* Financial support from the Canada Excellence Research Chair in Water Security and the Natural Science and Engineering Research Council's Changing Cold Regions Network is gratefully acknowledged. We also acknowledge support from Nordforsk Nordic eScience Globalisation Initiative (NeGI) project 74306 "An open-access generic e-platform for environmental model-building at the river-basin scale".

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

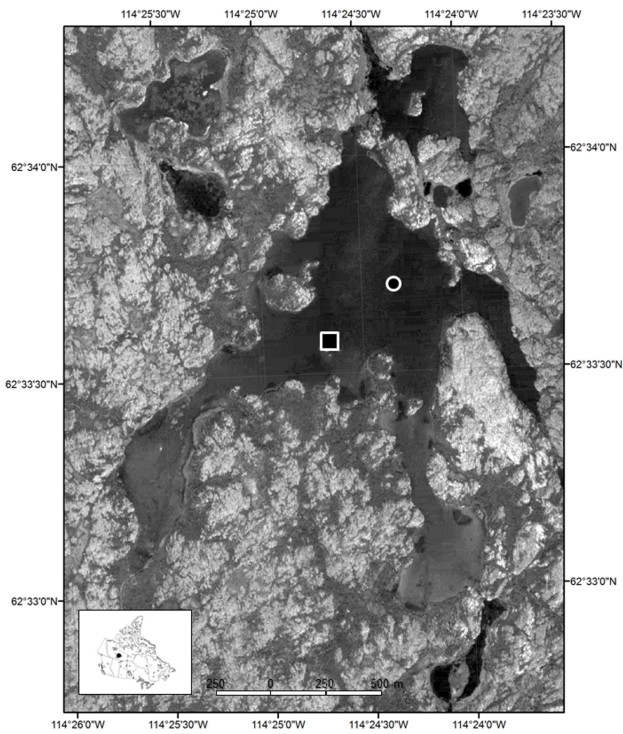

**Figure 1.** Aerial picture of Landing lake, with inset map indicating location within Canada. The black square and circle denote locations of climate tower and thermistor string respectively.

**Table 1.** Parameters of the Canadian Small Lake Model and their sampling ranges

| Parameter | | Abbreviation | Unit | Range |
|---|---|---|---|---|
| Extinction coefficient | | $K_d$ | m$^{-1}$ | $]0, 5[$ |
| | Wind | $c_n$ | | $]0.2, 2[$ |
| | Transport | $c_f$ | | $]0, 1[$ |
| TKE[a] | Dissipation | $c_e$ | | $]0, 1.7[$ |
| | Shear | $c_s$ | | $]0.2, 0.5[$ |
| | Leakage | $c_L$ | | $]0, 0.4[$ |

[a] Turbulent Kinetic Energy Budget

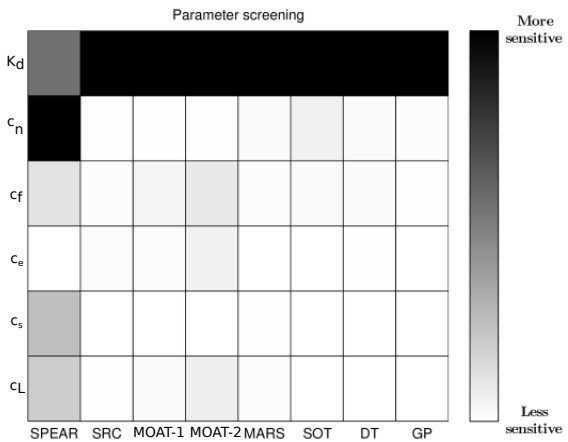

**Figure 2.** Comparison of the relative sensitivity of a subset of the parameters of the Canadian Small Lake Model, using the Nash-Sutcliffe performance to evaluate the model.

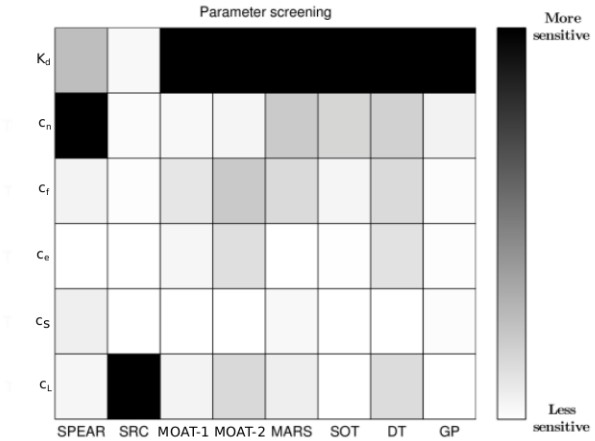

**Figure 3.** Comparison of the relative sensitivity of a subset of the parameters of the Canadian Small Lake Model, using the Mean Average Error to evaluate the model.

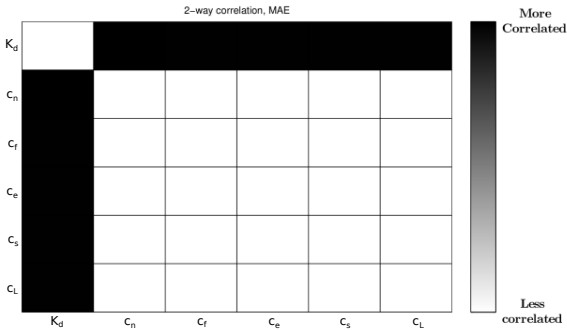

**Figure 4.** Two-way correlation between selected parameters of the Canadian Small Lake Model. This indicates how much the value of a given parameter is dependent on the value of another one to produce good simulations. The Mean Absolute Error was used to evaluate the model. McKay-2 (McKay et al., 1999) was used to evaluate the correlation.

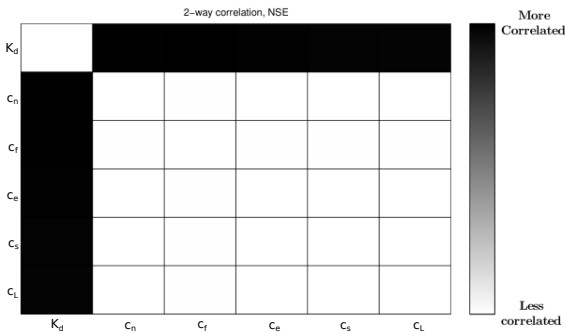

**Figure 5.** Two-way correlation between selected parameters of the Canadian Small Lake Model. This indicates how much the value of a given parameter is dependent on the value of another one to produce good simulations. The Nash-Sutcliffe performance was used to evaluate the model. McKay-2 (McKay et al., 1999) was used to evaluate the correlation.

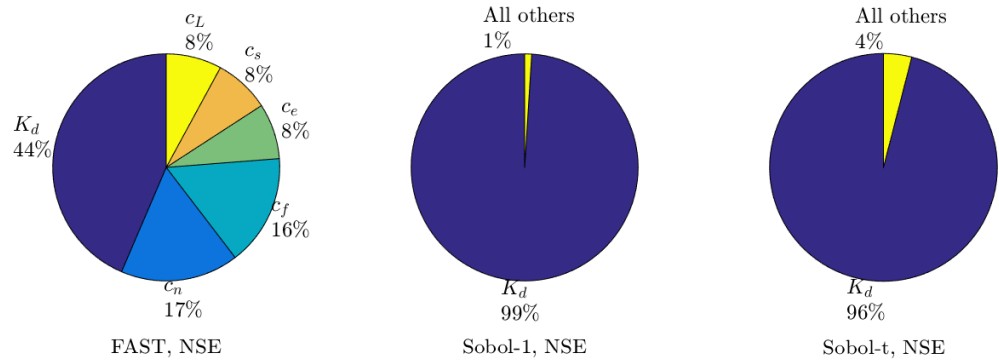

**Figure 6.** First and total order effects indicating the sensitivity of the model parameters on model performance. First order effects account for parameters independently whereas second order effects take include possible correlations between parameters. First order effects were computed with the FAST and Sobol-1 methods. Second order effects were computed using the Sobol-t method. The Nash-Sutcliffe performance was used to evaluate the model.

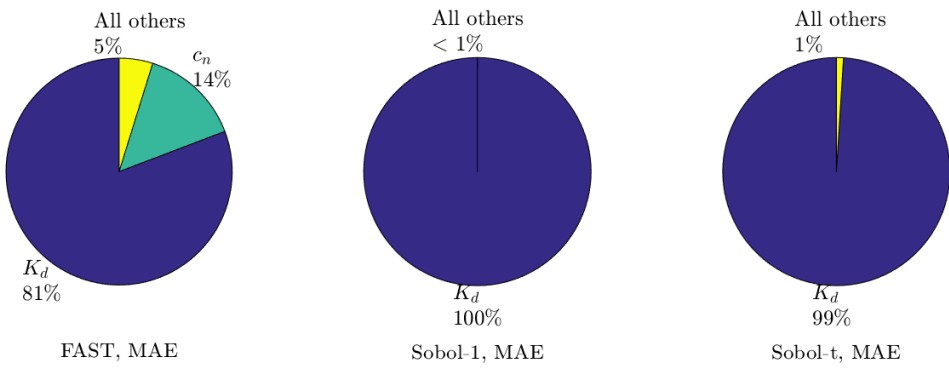

**Figure 7.** First and total order effects indicating the sensitivity of the model parameters on model performance. First order effects account for parameters independently whereas second order effects take include possible correlations between parameters. First order effects were computed with the FAST and Sobol-1 methods. Second order effects were computed using the Sobol-t method. The Mean Absolute Error was used to evaluate the model.

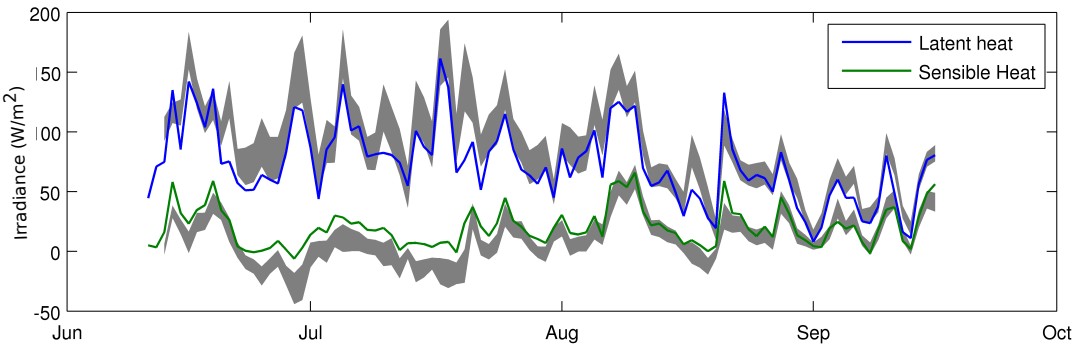

**Figure 8.** 5% and 95% uncertainty bounds for the heat fluxes, obtained from the GLUE global sensitivity method. The Nash-Sutcliffe performance was used to evaluate the model for the 2007 open-water season

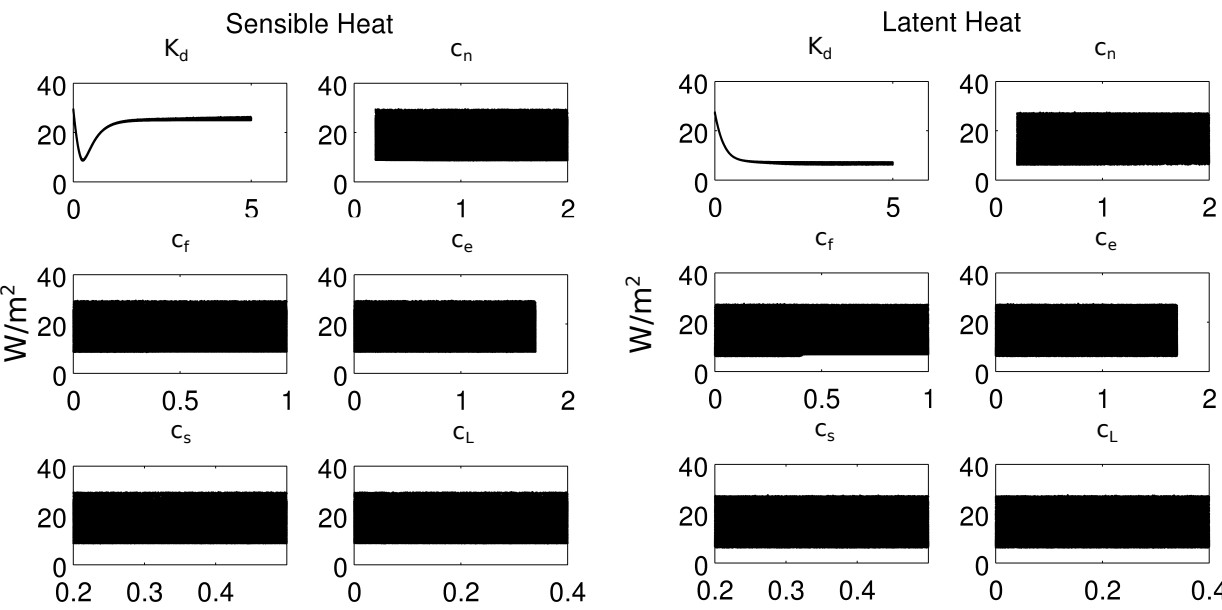

**Figure 9.** Mean Absolute Error for sensible and latent heat fluxes, from the GLUE simulations. Each dot represents a simulation with random parameters. Y-axis shows heat-flux in $W \cdot m^{-2}$ and x-axis is the parameter value. Results were similar for the Nash-Sutcliffe performance.

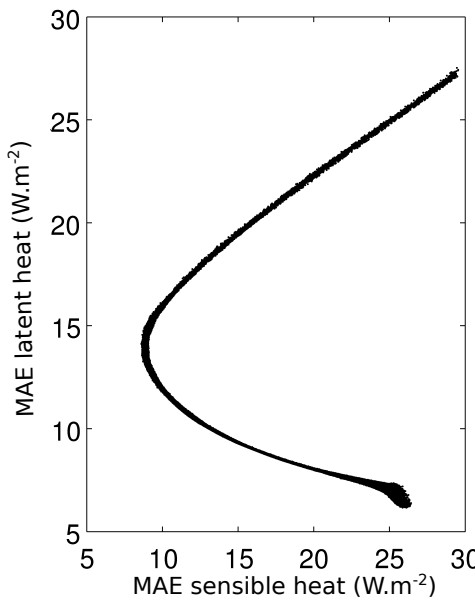

**Figure 10.** Latent vs sensible heat $W \cdot m^{-2}$ for behavioral (well-performing) simulations. Each dot was the output from a random simulation. The Mean Absolute Error was used to evaluate the model. Results were similar for the Nash-Sutcliffe performance.

**Table 2.** Sensitivity Analysis methods. Qualitative methods in black font. Quantitative methods in bold font.

| SA method | Abbreviation | Source |
|---|---|---|
| Correlation Analysis | SPEAR | Spearman (1904) |
| Regression Analysis | SRC | Galton (1886) |
| Morris-one-at-a-time screening | MOAT | Morris (1991) |
| Sum-of-trees screening | SOT | Breiman et al.(1984) |
| Gaussian process screening | GP | Gibbs and Mackay (1997) |
| Multivariate adaptive regression splines screening | MARS | Friedman (1991) |
| Delta-test screening | DT | Pi and Peterson (1994) |
| **Fourier amplitude sensitivity test** | **FAST** | **Cukier et al. (1973)** |
| **Sobol sensitivity indices** | **SOBOL** | **Sobol (1990, 2001)** |

**Table 3.** Setup for SA. Qualitative methods in black font. Quantitative methods in bold font.

| SA method | Sampling technique | Sample Size |
|---|---|---|
| SPEAR | MC | 10000 |
| SRC | MC | 10000 |
| MOAT | MOAT | 10000 |
| SOT | METIS | 3000 |
| GP | METIS | 3000 |
| MARS | METIS | 3000 |
| DT | METIS | 3000 |
| **FAST** | **FAST** | **373** |
| **SOBOL** | **SOBOL** | **3000** |

Sampling techniques:

a) MC: Uniform Sampling

b) MOAT: Designed specifically for MOAT

c) METIS: Space filling method

d) FAST: Designed specifically for FAST

e) SOBOL: Designed specifically for SOBOL See Gan et al. (2014) for a description of the sampling methods.

**Table 4.** Parameter sensitivity rankings of different qualitative sensitivity analysis methods

| Parameter | Sensitivity measure | | | | | | | |
|---|---|---|---|---|---|---|---|---|
| | SPEAR | SRC | MOAT-1 | MOAT-2 | MARS | SOT | DT | GP |
| $K_d$ | -0.27 | 0.08 | 0.10 | 0.10 | 100.00 | 1.00 | 1.00 | 100.00 |
| $c_n$ | -0.04 | 0.02 | 0.01 | 0.01 | 10.00 | 0.06 | 0.27 | 5.66 |
| $c_f$ | -0.26 | -0.06 | 0.01 | 0.03 | 10.00 | 0.07 | 0.23 | 1.37 |
| $c_e$ | -0.00 | -0.01 | 0.00 | 0.02 | 3.00 | 0.01 | 0.24 | 0.14 |
| $c_s$ | -0.02 | -0.02 | 0.00 | 0.01 | 4.00 | 0.02 | 0.21 | 0.05 |
| $c_L$ | 0.11 | 0.03 | 0.01 | 0.02 | 5.00 | 0.03 | 0.22 | 0.46 |