# Peer review of "Parameter sensitivity analysis of a 1-D cold region lake model for land-surface schemes"

_Hydrology and Earth System Sciences, 2017_

## Short Comment (SC1) · 16 Jan 2017

Dear,

It is a very nice study and important for understanding the role of lakes in the energy balance of cold regions. Just to make you aware that there are already several 1-D lake models for cold regions, especially the pan-Arctic, such as Stepanenko et al. (2011) and Tan et al. (2015).

Stepanenko, V. M., Machul'skaya, E. E., Glagolev, M. V., and Lykossov, V. N.: Numerical modeling of methane emissions from lakes in the permafrost zone, Izvestiya, Atmos. Oceanic Phys., 47(2), 252–264, 2011.

[Figure]

Tan, Z., Zhuang, Q., and Walter Anthony, K.: Modeling methane emissions from arctic lakes: Model development and site-level study, J. Adv. Model. Earth Syst., 7, 459–483, 2015.

Best regards

Zeli Tan (tan80@purdue.edu)

---

## Referee Comment (RC1) · Anonymous Referee #1 · 5 Feb 2017

The authors present an application of a third-party toolbox and the GLUE methodology to perform sensitivity and uncertainty assessments of heat exchange fluxes simulated by a 1D lake model during the open water season for a small lake in northern Canada. While some interesting material is presented, there is a lack of focus which makes it difficult to evaluate the usefulness of the results in a more general context.

In their parameter sensitivity analysis, the authors identify $K_d$, the light extinction coefficient, as the most important parameter controlling model performance. They then suggest that $K_d$ should be measured more widely as part of routine limnologic monitoring programs. While I agree with this sentiment, I am concerned that the authors do not present any measurements of $K_d$ but only note that the lake has "an expected $K_d$

value of ∼2m-1" (p.4 l.23). From the results presented, I can see that Kd is the most sensitive parameter controlling model performance, the failure of the authors to present Kd measurements and then argue for its widespread measurement is not logical and must be re-thought. It is possible that the lake has a markedly different measured Kd than that which resulted in the best model performance. If I am interpreting Figure 9 correctly, this is exactly what the authors show as the MAE for sensible heat flux is minimal when Kd is approximately 0.5. This discrepancy between expected Kd and simulated Kd leading to best model performance seriously undermines the results presented here and calls into question the overall validity of the modelling, suggesting that the authors have obtained the right results (i.e. a good fit to latent and sensible heat fluxes) for the wrong reasons (a model parameter Kd value of 0.5 when the authors expect the true value to be closer to 2.0).

On p.13 l.4-5, the authors note the all too common disconnect between experimentalists and modellers and suggest theirs is a contribution to addressing this problem. I am afraid the results presented which emphasize the importance of Kd for simulating heat transfer and then fail to remark on the disconnect between a hypothesized Kd value of around 2.0 and best model performance with a Kd value of approximately 0.5 only serve to highlight the deep and ongoing disconnection, even amongst co-authors on the same paper.

This difference in Kd values could serve as the starting point for an improved dialog between modeller and experimentalist. For example, what would the consequences have been for model performance if Kd had been fixed at 2.0, and under what circumstances would a hypothesized Kd of 0.5 have seemed reasonable? Throughout the manuscript, I am concerned that the authors are not aware of the relevant literature. For example, on p.1 l.14-15, the authors note that Kd is seldom measured. This is not entirely true, see e.g. Kalff (1992) , Ask et al. (1999), but secchi disk transparency and/or dissolved organic carbon are widely measured, and can be used to estimate Kd, an observation first published in 1929 for marine systems (Poole and Atkins 1929) and later refined

for lakes (Carlson 1977; Graneli et al. 1996). Furthermore, while Perez-Fuentetaja et al. (1999) and Tanetzap et al. (2008) are nice papers, I do not believe they are the best ones to support the authors' assertion that multiple processes operate in lakes at multiple time scales. This might be taken as a given from e.g. Kalff (2002).

The model description is inadequate to evaluate the significance of the findings presented. As the original model description paper is not open access, the authors must provide more detail in the present paper. Specifically, they need to provide a description of the manner in which Kd is used in model calculations. The authors also need to provide more detail about model execution. On what time scale and over what date range was the model run? It appears that field observations made at a 30 minute resolution were available. Was the model run on the same time step?

There is too little information provided about the empirical data collection. Over what time period were samples collected and what is the uncertainty in estimated heat fluxes? Using these uncertainties to inform model calibration and sensitivity analysis would have made for a much more informative paper than one which appears to compare modelled values to daily average flux estimates (as seems to be the case from Figure 8) using MAE and NS statistics.

The overview of sensitivity analysis needs to be rethought. In its present format, it is not sufficiently informative. P.6 l.25-26 makes an important and under-appreciated point, but apart from that, much of the text could be deleted and the reader referred to the more thorough discussions identified on p.5 l.23. The authors' description of PSUADE on p. 6 l.15-20 is inadequate. No indication is given as to code availability, language it is written in, etc.

I am of two minds about the description of sensitivity metrics. They are too short, but in light of the authors' subsequent findings, this may not matter as for the task at hand, they do not provide any real advance over older methods. The conclusion I draw from the authors' results is that sophisticated sensitivity analysis toolboxes such as

the PSUADE package they used are not needed for environmental modelling as one can derive the same information from an "old school" GLUE analysis. The dotty plots (if I am interpreting them correctly) suggest that Kd is the only sensitive parameter for both sensible and latent heat fluxes. It does not appear that application of the PSUADE package offers any additional insight above and beyond that obtained from the GLUE analysis. This, in and of itself, is a useful finding as it suggests researchers can concentrate on tried and true methods of sensitivity analysis instead of following the latest fads and fashions.

Minor Comments

P.1 authors – is there an error here and should the third author be Howard Wheater?

p.2 l.6-19 – This discussion of the manner in which lakes are incorporated into climate models is interesting but irrelevant to the authors' stated objectives of performing a sensitivity analysis. While the CSLM has been developed for climate change studies, this is outside the scope of its use in the current paper. Thus, I would ask that the authors delete or greatly shorten this section. Expanding upon the statement on p.2 l. 28-30 would provide more relevant background information.

p.3 l.23 – higher values of Kd do not necessarily indicate more turbid lakes. High dissolved organic carbon concentrations and an absence of turbidity can also result in high Kd.

p.6 l.27 – I dispute the authors' assertion that ". . .evaluation of heat fluxes over northern lakes remain uncommon . . .". I would encourage the authors to consult Rouse et al. (2005), if only to put their results into context.

Figures

Please replace Figure 1 with a bathymetric map of the lake showing the location of the thermistor arrays. This would help the reader to judge the statement made on p.5 l.10 and to better understand the relationship between mean and max depth presented on

[Figure]

p.4 l.21.

Figures 2-5 are not terribly useful. Please delete them as one can derive the same information from Table 4.

The information in Figures 6 and 7 could be presented more succinctly as a table.

Figure 8 is encouraging as it shows the model is able to reproduce the observations. I do have some concerns, however. Does Figure 8 present data for a single year? If so, which one? Please also provide some estimate of uncertainties in the latent and sensible heat fluxes.

I have to admit that figure 9 confuses me. I assume that the MAE has units of W/m2? If so, please clarify this in the figure caption. I would like to see a similar set of plots based on the NS statistic.

Figure 10 deserves more consideration in the paper. It is a really useful piece of information that there is a non-monotonic relationship between the MAEs for latent and sensible heat flux. I would strongly encourage the authors to explore how this looks when using the NS, also.

References

Ask, J., Karlsson, J., Persson, L., Ask, P., Byström, P. and Jansson, M., 2009. Terrestrial organic matter and light penetration: Effects on bacterial and primary production in lakes. Limnol. Oceanogr, 54(6), pp.2034-2040.

Carlson, R.E., 1977. A trophic state index for lakes. Limnol. Oceanogr., 22(2), pp.361-369.

Graneli, W., Lindell, M. and Tranvik, L., 1996. Photo-oxidative production of dissolved inorganic carbon in lakes of different humic content. Limnology and Oceanography, 41, pp.698-706.

Kalff, J., 2002. Limnology: inland water ecosystems (Vol. 592). New Jersey: Prentice

Hall.

Poole, H.H. and Atkins, W.R.G., 1929. Photo-electric measurements of submarine illumination throughout the year. Journal of the Marine Biological Association of the United Kingdom (New Series), 16(01), pp.297-324.

Rouse, W.R., Oswald, C.J., Binyamin, J., Spence, C., Schertzer, W.M., Blanken, P.D., Bussières, N. and Duguay, C.R., 2005. The role of northern lakes in a regional energy balance. Journal of Hydrometeorology, 6(3), pp.291-305.

———————————

---

## Referee Comment (RC2) · Anonymous Referee #2 · 20 Feb 2017

Review of 'Parameters sensitivity analysis of a 1-D cold region lake model for land-surface schemes' by Guerrero et al.

General comments This is a reasonably written paper describing an interesting topic in environmental modeling and numerical weather prediction: 'How do lakes interact with their overlying atmosphere and to what extent can lakes modify their surrounding climate, and the uncertainties in these interactions'. A number of previous papers have addressed similar topics in the past (e.g. Dutra et al. 2010; Balsamo et al. 2012), but the strength of this current paper is the uncertainty estimation that it provides. Specifically, the authors introduce a third-party toolbox and the GLUE methodology to perform a sensitivity and uncertainty analysis of the different surface heat fluxes

simulated by the Canadian Small Lake Model (CSLM), a one-dimensional integral lake model. The authors focus their study on a small lake in northern Canada, which is a good study site as small lakes are the most abundant at the global scale (see further my notes in the specific comments below). Within their sensitivity analysis, the authors find that the light attenuation coefficient, Kd, is the most important parameter controlling model performance and that variable Kd provides the highest uncertainty in surface flux estimates. I don't particularly find this surprising, as others have found that water clarity can have a considerable influence on lake stratification and the turbulent heat fluxes (see Heiskanen et al. 2015) and can also considerably influence the diurnal cycles of heating and cooling in lakes (Woolway et al. 2016), but I do find this an important point to highlight and one that deserves some attention. While I think this paper will be of interest to those who focus on the integration of lakes within the climate system and for Numerical Weather Prediction, I strongly believe that the paper would be improved if there were more focus on the analysis and the results were put into context of the published literature. Often I found some of the most relevant literature being ignored and/or overlooked and some references, which were included in the text, seem inappropriate or irrelevant. One of my main criticisms is that a thorough literature review is needed to strengthen the introduction and discussion of the results. I provide some examples of relevant studies in this review, but there are many others which the authors should also look into. I strongly suggest a thorough review of the current literature prior to publication. I find it surprising that the authors specify that Kd is the most important parameter controlling model performance, but do not include any detailed measurements of Kd. In particular, it is very likely that the lake has a different Kd to that estimated from the model sensitivity analysis. Overall, I think there is some potential for this paper to be revised sufficiently to make it a valuable contribution to the scientific literature. However, addressing all of the points raised below are needed, in my opinion, prior to this paper being considered for publication in HESS.

Specific Comments Unfortunately the Downing et al. (2006) estimates of global lake size and abundance are no longer supported. Many studies have since shown that

the Pareto distribution does not adequately describe the global distribution of lakes. For example, see Seekell and Pace (2011) and McDonald et al. (2012). A more detailed description of the global abundance and size distribution of lakes are provided by Verpoorter et al. (2014) and more recently by Cael and Seekell (2016). Granted that these recent studies to do not consider the smallest lakes of the world (for example, Verpoorter et al. only consider lakes larger than 0.002 km2), but still the authors should read up on these papers and include the relevant citations.

'They also provide a more immediate feedback through mass and energy exchanges with the atmosphere' - you need some reference for this. As I'm sure you're aware, these fluxes are quite difficult to calculate (see Woolway et al. 2015a). Further information on these fluxes is needed, in my opinion. Additional information here will allow others who are not experts in the field to understand better the kind of interactions you are talking bout.

Tanentzap et al. (2008) did not consider the influence of variations in thermocline depth on fluxes to the atmosphere, thus I don't think this reference is appropriate.

'Rinke et al. (2010) illustrate the feedback between phytoplankton and thermal structure...' - There are other studies which you could also cite. For example, Mazumder et al. (1990) showed this over two decades ago. There are many other studies since then which I think the authors should read up on.

It may also be worth mentioning that, on a regional scale, Samuelsson et al. (2010) found that the presence of lakes induces a warming on the European climate, and an observational study by Rouse et al. (2005) found that high-latitude lakes strongly enhance evapotranspiration when added to a landscape. A useful study, which I think the authors should cite, is Heiskanen et al. (2015). The authors should also look at the papers cited by Heiskanen et al. (2015) as these will be of direct relevant to this study. In addition, a paper by Rose et al. (2016) describes that water clarity can either amplify or suppress lake surface water temperatures, which in turn will influence their

interaction with the atmosphere. Please read the Rose et al. (2016) paper and look at the references within.

A lake depth sensitivity analysis was undertaken by Balsamo et al. (2010) and might be worth mentioning also.

P2L26 - What is a small lake? How do you characterize a lake as small?

P3L24 - Water clarity can have numerous other influences on lake temperatures. I think this section needs to be expanded. A few examples include its influence on the thermal structure of lakes (e.g. Persson and Jones 2008), its influence on the absorption of heat during the day and greater release in the evening leading to larger diurnal cycles (Woolway et al. 2015b) and influencing the likelihood of diurnal stratification as well as seasonal stratification. Also, studies have shown that surface waters have been browning over the last few decades (Roulet and Moore 2006). All of these points should be included and expanded.

Italics aren't needed for the description of all units.

P12L30 - The authors state that Kd is not often measured and measuring Kd for every lake might be a practical impossibility. In my opinion, this is one of the largest uncertainties in the inclusion of lakes in NWP. For example, in ECMWF's IFS Kd is assumed equal to 3 for all lakes, which could result in numerous biases in the turbulent heat fluxes. While I somewhat agree with the author's statements here, it may also be worth mentioning that satellites can estimate Kd, so there are possibilities in improving lake surface water temperature simulations. For more information, see Torbick et al. (2013) for information on how satellites can potentially be used to estimate secchi depth, which can be used as an indicator of Kd.

P13L3 - 'this kind of monitoring has never been performed' - This isn't true. Lake monitoring stations now often have light sensors above and below the water surface and are thus used to determine water clarity and Secchi depth observations are traditionally recorded. I suggest the authors look through the literature to find examples of where they've been used. I'm almost certain that this information has not been used in NWP or climate modeling, but I hope in the future meteorologists and limnologists will work closer to address this and similar issues. A literature search on this topic is also needed in my opinion.

P13L9 - I don't think this can be a main conclusion as unfortunately it is not unknown. For example, see Heiskanen et al. (2015).

There doesn't appear to be much discussion in this paper. I would recommend restructuring the paper to include separate 'Results' and 'Discussion' sections and perhaps reduce the conclusion to one or two paragraphs. This, in my opinion, would make the paper easier to digest.

I don't find many of the figures presented in the paper very informative. They seem to all show similar results. Much of this information could be shown in 1 or 2 figures, in my opinion.

Figure 1 needs more information. For example, can the authors add a smaller inset map to show where the lake is? Also, the figure would need a 'scale ruler' so that the reader can easily interpret the size of the lake.

At first glance, I don't particularly understand Fig. 8. It isn't clear what the grey regions represent as one would expect the grey area to be an envelope that surrounds the main (blue) line?

Fig. 9 - Isn't irradiance a term often used to describe solar irradiance and not the turbulent fluxes? Also, why isn't there an x-label on the bottom panels?

Fig. 10 - I'm not sure how to interpret this figure. Can you please provide a better description of what we're seeing? I think a more detailed discussion of this figure should be given in the text.

Table 1 - The square brackets appear the wrong way round in the fourth column.

References: Balsamo G, et al (2010), Deriving an effective lake depth from satellite lake surface temperature data: a feasibility study with MODIS data. Boreal Environment Research 15:178-190.

Balsamo G, Salgado R, Dutra E, Boussetta S, Stockdale T, Potes M (2012), On the contribution of lakes in predicting near-surface temperature in a global weather forecasting model. Tellus A 64, 15829.

Cael BB, Seekell DA (2016), The size-distribution of Earth's lakes. Sci Rep 6, 29633.

Dutra E, Stepanenko VM, Balsamo G, Viterbo P, Miranda PM, et al. (2010), An offline study of the impact of lakes on the performance of the ECMWF surface scheme. Boreal Env. Res. 15:100–112.

Heiskanen JJ, et al. (2015), Effects of water clarity on lake stratification and lake-atmosphere heat exchange. J Geophys Res Atmos 120:7412-7428

Mazumder A, Taylor WD, McQueen DJ, Lean DR (1990), Effects of fish and plankton and lake temperature and mixing depth. Science 247:312–315.

McDonald CP, et al. (2012), The regional abundance and size distribution of lakes and reservoirs in the United States and implications for estimates of global lake extent. Limnol. Oceanogr. 57:597-606.

Persson I, Jones ID (2008) The effect of water colour on lake hydrodynamics: a modeling study. Freshwater Biol 53:2345-2355

Rose KC, Winslow LA, Read JS, Hansen GJA (2016) Climate-induced warming of lakes can be either amplified or suppressed by trends in water clarity. Limnol Oceanogr Lett 1:44-53.

Roulet N, Moore TR (2006) Environmental chemistry: Browning the waters. Nature 444:283–284.

Seekell DA, Pace ML (2011), Does the pareto distribution adequately describe the
size-distribution of lakes? Limnol. Oceanogr. 56(1):350-356.

Torbick N, Hession S, Hagen S, Wiangwang N, Becker B, Qi J (2013) Mapping inland lake water quality across the Lower Peninsula of Michigan using Landsat TM imagery. Int J Remote Sens, 34:7607–7624.

Verpoorter C, Kutser T, Seekell DA, Tranvik LJ (2014), A global inventory of lakes based on high-resolution satellite imagery. Geophys Res Lett 41:6396-6402.

Woolway, R.I, Jones, I.D., Hamilton, D.P. et al. (2015a). Automated calculation of surface energy fluxes with high-frequency lake buoy data. Environmental Modelling & Software 70, 191-198.

Woolway, R.I., Jones, I.D., Feuchtmayr, H. et al. (2015b). A comparison of the diel variability in epilimnetic temperature for five lakes in the English Lake District. Inland Waters 5(2), 139-154.

Woolway, R.I., Jones, I.D., Maberly, S.C. et al. (2016). Diel surface temperature range scales with lake size. PLoS One 11(3): e0152466. doi: 10.1371/journal.pone.0152466

---

## Author Comment (AC2) · 11 Apr 2017

Please refer to the reply to the short comment, where we reply to all comments.

---

## Author Comment (AC3) · 11 Apr 2017

Please refer to the reply to the short comment, where we reply to all comments.

---

## Editor Comment (EC1) · D. Solomatine (Editor) · 10 May 2017

It was an interesting discussion, enthusiastic (and at places quite critical) comments from referees and replies from the authors. I am inviting the authors to update the manuscript accordingly, and to submit it together with rebuttal (based on already published replies).

I would like to add one thing, for possible consideration: Figure 9 (results of GLUE) shows that MAE (and hence model output) does not depend on values of Cf and Cs (and it is also stated on p 12). How physical is this? How does this Figure relate to Figures 6-7 where one can see that MAE or NSE are indeed sensitive to Cs and Cf?

---

## Author Response (AR1)

**Reply to comments on "Parameter sensitivity analysis of 1-d cold region lake model for land-surface schemes" by José-Luis Guerrero et al.**

We'd like to start by thanking Zeli Tan and two anonymous reviewers for the time they spent going through this work and providing feedback. We also would like to than the editorial office and the editor for the prompt handling of this paper. In this document we address the reviewers' concerns. The original comments are in boldface and our answers in plain text.

Following editorial input, our modifications to the originally submitted text are highlighted in blue font in this document. The page and line numbers correspond to the updated manuscript. We also added a reply to the editor's comments.

**Reviewer Z. Tan, short comment**

**Dear,**

**It is a very nice study and important for understanding the role of lakes in the energy balance of cold regions. Just to make you aware that there are already several 1-D lake models for cold regions, especially the pan-Arctic, such as Stepanenko et al. (2011) and Tan et al. (2015).**

**Stepanenko, V. M., Machul'skaya, E. E., Glagolev, M. V., and Lykossov, V. N.: Numerical modeling of methane emissions from lakes in the permafrost zone, Izvestiya, Atmos. Oceanic Phys., 47(2), 252–264, 2011.**

**Tan, Z., Zhuang, Q., and Walter Anthony, K.: Modeling methane emissions from arctic lakes: Model development and site-level study, J. Adv. Model. Earth Syst., 7, 459–483, 2015.**

**Best regards Zeli Tan (tan80@purdue.edu)**

Thank you for the kind comments. Here we focused on energy feedbacks. Greenhouse gases such as methane are of course important in the climate system and should eventually be included in largescale modeling systems. The provided literature could be cited as examples of other 1-D models.

Changed p.2, l.28: *"[...]who show how lakes impact regional climate."* to

*"[...] who show how lakes impact regional climate and contribute to greenhouse gas emissions (Stepanenko et al., 2011; Tan et al., 2015). "*

**Anonymous reviewer #1.**

**The authors present an application of a third-party toolbox and the GLUE methodology to perform sensitivity and uncertainty assessments of heat exchange fluxes simulated by a 1D lake model during the open water season for a small lake in northern Canada. While some interesting material is presented, there is a lack of focus which makes it difficult to evaluate the usefulness of the results in a more general context.**

Thank you for your input and we are glad you found some of the material interesting. We will attempt in the replies to the rest of your comments to clarify the focus and usefulness and the paper.

Modifications of the text will be undertaken upon editorial input, as per HESS guidelines.

he background behind the inception of this paper was to use the relatively scarce latent and heat fluxes modeling data to improve lake modeling within land-surface schemes. This led us to stress the importance of Kd for good modeling results, showing that a wide array of sensitivity analysis methods agree on this result.

**In their parameter sensitivity analysis, the authors identify Kd, the light extinction coefficient, as the most important parameter controlling model performance. They then suggest that Kd should be measured more widely as part of routine limnologic monitoring programs. While I agree with this sentiment, I am concerned that the authors do not present any measurements of Kd but only note that the lake has "an expected Kd value of ∼2m-1" (p.4 l.23).**

And this is precisely the problem: oftentimes Kd is not measured. We were not involved in the field measurement campaign and only made use of the final product. Our point is that Kd should be a routine measurement in all such campaigns, a continuous measurement if possible.

**From the results presented, I can see that Kd is the most sensitive parameter controlling model performance, the failure of the authors to present Kd measurements and then argue for its widespread measurement is not logical and must be re-thought. It is possible that the lake has a markedly different measured Kd than that which resulted in the best model performance.**

Please refer to our previous comment. We only have an approximate value of Kd, which does not even take into account the temporal variability.

**If I am interpreting Figure 9 correctly, this is exactly what the authors show as the MAE for sensible heat flux is minimal when Kd is approximately 0.5. This discrepancy between expected Kd and simulated Kd leading to best model performance seriously undermines the results presented here and calls into question the overall validity of the modelling, suggesting that the authors have obtained the right results (i.e. a good fit to latent and sensible heat fluxes) for the wrong reasons (a model parameter Kd value of 0.5 when the authors expect the true value to be closer to 2.0).**

A clearer explanation of the dotty plots will be provided. What Fig. 9 shows is that MAE is minimal when Kd is around 0.5.

We do not believe this undermines our results. We might in fact argue that this is the curse of most, if not all, models: equifinality and the disconnect between model parameters and the variable they are supposed to represent.

In this case, the divergence can be explained, e.g. by a possible temporal variability in the value of Kd.

Even if we accept that the measurement was accurate at the time it was taken, this did not coincide with our modeling period and the light attenuation might vary over the course of a season so the parameter in the model is sort of like an "average".

If Kd was actually measured, it could with confidence be set as a constant (perhaps time-dependent) in the model as opposed to be considered a calibration parameter.

Added (p.10, l.28): To each parameter values combination corresponds a performance and the projection of the (multidimensional) parameter-values against performance along one dimension, or parameter-axis, produces what are commonly known as ``dotty plots'' and can give an idea of parametric sensitivity (Beven, 2006).

**On p.13 l.4-5, the authors note the all too common disconnect between experimentalists and modellers and suggest theirs is a contribution to addressing this problem. I am afraid the results presented which emphasize the importance of Kd for simulating heat transfer and then fail to remark on the disconnect between a hypothesized Kd value of around 2.0 and best model performance with a Kd value of approximately 0.5 only serve to highlight the deep and ongoing disconnection, even amongst co-authors on the same paper.**

Our contribution to this dialog could be resumed as follows: "please measure Kd continuously, this would allow furthering and improving modeling efforts". Please let us insist on the fact that we did not take the measurements ourselves but only made use of them. This should be explicitly stated in the paper.

Added (p.14, l.13): Modified "*The conceptual framework behind the present study is a testament to an improvement of those dialogues: the undertaken modeling approach was based on novel data from an observation station that was established explicitly to support testing of hydrometorological models over lakes. Such new data facilitated the analysis performed here.*" to

"*[...] explicitly established to support testing of hydrometeorological models over lakes:. Such new data facilitated the analysis performed here and it must be stressed that measuring Kd was not part of the objectives of the field campaign. The modeling exercise performed here underlined its importance for the simulation of heat fluxes and represents an argument in favor of its monitoring.*"

**This difference in Kd values could serve as the starting point for an improved dialog between modeller and experimentalist. For example, what would the consequences have been for model performance if Kd had been fixed at 2.0, and under what circumstances would a hypothesized Kd of 0.5 have seemed reasonable? Throughout the manuscript, I am concerned that the authors are not aware of the relevant literature. For example, on p.1 l.14-15, the authors note that Kd is seldom measured. This is not entirely true, see e.g. Kalff (1992) , Ask et al. (1999), but secchi disk transparency and/or dissolved organic carbon are widely measured, and can be used to estimate Kd, an observation first published in 1929 for marine systems (Poole and Atkins 1929) and later refined for lakes (Carlson 1977; Graneli et al. 1996). Furthermore, while Perez-Fuentetaja et al. (1999) and Tanetzap et al. (2008) are nice papers, I do not believe they are the best ones to support the authors' assertion that multiple processes operate in lakes at multiple time scales. This might be taken as a given from e.g. Kalff (2002).**

The reviewer is undeniably right and these references should be added and some of our assertions modified accordingly.

Modified p.1, l.14 from: *"This is important since Kd is seldom measured and accurate estimates of its value could reduce modeling uncertainty."* to *"This is important since accurate and continuous measurements of Kd could reduce modeling uncertainty."*

p1, l.22: Added Kalff(2002) as a citation to support the assertion of different processes working at different time scales.

Added text, p.5, l.2: *"Although measurement technology has long been available (Poole and Atkins, 1929), continuous measurements of K d are, to the best of our knowledge, relatively scarce despite the relative affordability of cosine collectors, perhaps currently the most widespread technology; see Frankovich et al. (2017) for a recent application. Also, point measurements can be taken through more or less direct proxies, such as the Secchi disk depth (Tyler, 1968) and Dissolved Organic Compounds (DOC) concentrations (Ask et al., 2009). The technology is in fact evolving (Chudyk and Flynn, 2015). Alas, despite the availability of measurement tools, K d might sometimes be an afterthought, as in the case here presented, where it was determined from DOC concentrations."*

**The model description is inadequate to evaluate the significance of the findings presented. As the original model description paper is not open access, the authors must provide more detail in the present paper. Specifically, they need to provide a description of the manner in which Kd is used in model calculations. The authors also need to provide more detail about model execution. On what time scale and over what date range was the model run? It appears that field observations made at a 30 minute resolution were available. Was the model run on the same time step?**

While not open-access, it is still a part of the literature and we did not feel it was required to retake the detailed description already available, albeit behind a paywall. We whole-heartedly agree on the importance of open access, but this is perhaps not the best forum for the issue. We agree that we should provide more details on how Kd affects model calculations and more detail about model execution.

The model was run on a 30 min time step and the results aggregated to an daily time step due to inherent limitation of the eddy covariance data.

p.11, l.3: Changed *"Daily flux data, obtained through integration of hourly eddy-covariance measurements, were available for the period 12/06/2007–18/10/2007 and were compared to simulated fluxes to calculate two performance metrics, MAE and NSE:".*

*"The model was run with half-hourly forcings and the resulting simulations were temporally aggregated to evaluate against daily flux data, which was obtained through integration of hourly eddy-covariance measurements. The aggregation was necessary because of inherent limitations of the higher frequency data, that were available for the period 12/06/2007–18/10/2007. Two performance metrics were used for the evaluation, MAE and NSE: "*

Added description of Kd's function in the model, p.4, l.6 : " Within CSLM, shortwave extinction is exponential with depth following Beer's law, with $K_d$ identified as the e-folding depth."

**There is too little information provided about the empirical data collection. Over what time period were samples collected and what is the uncertainty in estimated heat fluxes? Using**

**these uncertainties to inform model calibration and sensitivity analysis would have made for a much more informative paper than one which appears to compare modeled values to daily average flux estimates (as seems to be the case from Figure 8) using MAE and NS statistics.**

We will clarify data description. There are many ways of tackling uncertainty analysis. Using the uncertainty in measurements to assess modeling uncertainty is known as the limits-of acceptability approach. This was not our objective here but should indeed be the basis of a more robust approach to uncertainty evaluation, in our opinion.

For data clarification, please refer to the reply to the previous comment.

**The overview of sensitivity analysis needs to be rethought. In its present format, it is not sufficiently informative. P.6 l.25-26 makes an important and under-appreciated point, but apart from that, much of the text could be deleted and the reader referred to the more thorough discussions identified on p.5 l.23. The authors' description of PSUADE on p. 6 l.15-20 is inadequate. No indication is given as to code availability, language it is written in, etc.**

This section as originally written, was much more technical in the description. Before submission it was decided to give a more conceptual overview of the different methods instead of focusing on the technical details, which would have yielded a much longer and perhaps murkier paper.

To address the reviewer's concern we could better highlight which of the cited papers provide detailed descriptions of the methods used. Please also note that table 2 provides the original sources for the methods used.

PSUADE is an open-source package written in C++. This should be made clearer

Added p.6, l.12 : "Song (2015) in particular provides an exhaustive overview of the state-of-the art."

p.1, l.6. Changed "*A recently developed software package*" to " *A recently developed C++ open-source software package*"

**I am of two minds about the description of sensitivity metrics. They are too short, but in light of the authors' subsequent findings, this may not matter as for the task at hand, they do not provide any real advance over older methods. The conclusion I draw from the authors' results is that sophisticated sensitivity analysis toolboxes such asthe PSUADE package they used are not needed for environmental modelling as one can derive the same information from an "old school" GLUE analysis.**

The reviewer is right in it was not our intention to advance existing methods but to apply them.

We strongly disagree that packages such as PSUADE are not needed for environmental modeling. On one hand, different methods make different assumptions that may be more or less warranted and in that sense, to have an entire array of methods available con only be an advantage.

Also, taking only practical considerations into account, the choice of an SA method might boil down to familiarity with these methods and having an array of methods under one roof can help overcome this hurdle through a united framework.

Also, GLUE is computationally demanding and for other models might it might be a practical impossibility to use it.

**The dotty plots (if I am interpreting them correctly) suggest that Kd is the only sensitive parameter for both sensible and latent heat fluxes. It does not appear that application of the PSUADE package offers any additional insight above and beyond that obtained from the GLUE analysis. This, in and of itself, is a useful finding as it suggests researchers can concentrate on tried and true methods of sensitivity analysis instead of following the latest fads and fashions.**

We would not say that the SA methods do not provide additional insight, but that they agree with the GLUE results and the results are more robust as a consequence of this since the different methods are based on different assumptions, as described in the Methods section.

**Minor Comments**

**P.1 authors – is there an error here and should the third author be Howard Wheater?**

Thank you for pointing out the typo.

Fixed

**p.2 l.6-19 – This discussion of the manner in which lakes are incorporated into climate models is interesting but irrelevant to the authors' stated objectives of performing a sensitivity analysis. While the CSLM has been developed for climate change studies, this is outside the scope of its use in the current paper. Thus, I would ask that the authors delete or greatly shorten this section. Expanding upon the statement on p.2 l. 28-30 would provide more relevant background information.**

We felt it should be mentioned that this paper came to be as part of an effort of improving land-surface schemes, although we agree this has no direct impact on our narrative. We feel that this section provides necessary background on the evolution of lake modeling of which the CSLM is an example of currently existing approaches. We can expand on fluxes over lakes (p2,l28-30)

Expanded upon the statement with p.3, l.1: *"With different albedo, heat capacity and surface roughness than the surrounding land areas lakes also provide more immediate feedback through transfer of heat and moisture exchanges with the atmosphere (e.g., MacKay et al., 2009; Xiao et al., 2013; McGloin et al., 2014b). While some studies have performed direct measurements of latent and sensible turbulent heat fluxes from eddy covariance systems over lakes and reservoirs (e.g., Blanken et al., 2000; Vesala et al., 2006; Blanken et al., 2011; Nordbo et al., 2011; McGloin et al., 2014a) these measurements can be difficult and expensive and as such improved modelling approaches are necessary (e.g., McGloin et al., 2014b)."*

**p.3 l.23 – higher values of Kd do not necessarily indicate more turbid lakes. High dissolved organic carbon concentrations and an absence of turbidity can also result in high Kd.**

The reviewer is right. Algal blooms will also change Kd. A better discussion of this should be provided.

Please refer to the added text for a previous comment:

Added text, p.5, l.2: *"Although measurement technology has long been available (Poole and Atkins, 1929),continuous measurements of K d are, to the best of our knowledge, relatively scarce despite the relative affordability of cosine collectors, perhaps currently the most widespread technology; see Frankovich et al. (2017) for a recent application. Also, point measurements can be taken through more or less direct proxies, such as the Secchi disk depth (Tyler, 1968) and Dissolved Organic Compounds (DOC) concentrations (Ask et al., 2009). The technology is in fact evolving (Chudyk and Flynn, 2015). Alas, despite the availability of measurement tools, K d might sometimes be an afterthought, as in the case here presented, where it was determined from DOC concentrations."*

**p.6 l.27 – I dispute the authors' assertion that ". . .evaluation of heat fluxes over northern lakes remain uncommon . . .". I would encourage the authors to consult Rouse et al. (2005), if only to put their results into context.**

Changed the text to read, p.7, l.15: *"The present study was based on a combination of three factors underlining its relevance. Firstly, by building upon existing literature that stresses the importance of lake clarity in modeling heat transfers (Heiskanen et al., 2015; Rose et al., 2016, Woolway et al., 2016), evaluating against measured fluxes, as done by (Deacu et al., 2012) for large lakes."*

**Figures**

**Please replace Figure 1 with a bathymetric map of the lake showing the location of the thermistor arrays. This would help the reader to judge the statement made on p.5 l.10 and to better understand the relationship between mean and max depth presented onFigures 2-5 are not terribly useful. Please delete them as one can derive the same information from Table 4.**

A revised figure could be provided.

Done.

Also replaced, p.5, l.11: "Landing Lake is a small freshwater lake with a surface area of 1.12 km2. mean depth of 3 m and maximum depth of 4 m" with

"Landing Lake is a small freshwater lake with a surface area of 1.12 km2). While no comprehensive bathymetry measurements have been taken on Landing Lake, depths in the main body of the lake during installations of thermistors and pressure transducers over the course of this study and others are consistently 4 m. The lake's two southern arms are shallower, near 1.5 m, as can be seen by the change in colouration in Fig. 1".

**The information in Figures 6 and 7 could be presented more succinctly as a table.**

We appreciated the visual impact of the shown figures to highlight the importance of Kd.

**Figure 8 is encouraging as it shows the model is able to reproduce the observations. I do have some concerns, however. Does Figure 8 present data for a single year? If so, which one? Please also provide some estimate of uncertainties in the latent and sensible heat fluxes.**

Results for 2007 are presented. This should be explicitly stated in the caption.

Fixed caption

The estimate of the uncertainties can be read from Fig.9. There is a trade-off in the modelling of the fluxes.

**I have to admit that figure 9 confuses me. I assume that the MAE has units of W/m2? If so, please clarify this in the figure caption. I would like to see a similar set of plots based on the NS statistic.**

We did not include a plot for NSE since the results were very similar. This should be explicitly stated.

Fixed caption. Clarified similarity to NSE in caption

**Figure 10 deserves more consideration in the paper. It is a really useful piece of information that there is a non-monotonic relationship between the MAEs for latent and sensible heat flux. I would strongly encourage the authors to explore how this looks when using the NS, also.**

We should expand the results sections to highlight this.

Clarified similarity to NSE in caption.

Added text to the conclusions, p.14, l.26: *"The clear tradeoffs inperformance for latent and sensible heat might be influenced by the fact while the CSLM surface energy balance is a strongly nonlinear function of the surface skin temperature, both the sensible and latent heat fluxes are linear terms in this relationship. All other things being equal, this leads to a direct tradeoff between them: the capacity of the model to simulate one of the terms is inversely proportional to its ability to simulate the other, see Fig. 10."*

**References**

**Ask, J., Karlsson, J., Persson, L., Ask, P., Byström, P. and Jansson, M., 2009. Terrestrial organic matter and light penetration: Effects on bacterial and primary production in lakes. Limnol. Oceanogr, 54(6), pp.2034-2040.**

**Carlson, R.E., 1977. A trophic state index for lakes. Limnol. Oceanogr., 22(2), pp.361- 369.**

**Graneli, W., Lindell, M. and Tranvik, L., 1996. Photo-oxidative production of dissolved inorganic carbon in lakes of different humic content. Limnology and Oceanography, 41, pp.698-706.**

**Kalff, J., 2002. Limnology: inland water ecosystems (Vol. 592). New Jersey: Prentice Hall.**

**Poole, H.H. and Atkins, W.R.G., 1929. Photo-electric measurements of submarine illumination throughout the year. Journal of the Marine Biological Association of the United Kingdom (New Series), 16(01), pp.297-324.**

**Rouse, W.R., Oswald, C.J., Binyamin, J., Spence, C., Schertzer, W.M., Blanken, P.D., Bussières, N. and Duguay, C.R., 2005. The role of northern lakes in a regional energy balance. Journal of Hydrometeorology, 6(3), pp.291-305.**

**Anonymous reviewer #2.**

**Review of 'Parameters sensitivity analysis of a 1-D cold region lake model for landsurface schemes' by Guerrero et al.**

**General comments**

**This is a reasonably written paper describing an interesting topic in environmental modeling and numerical weather prediction: 'How do lakes interact with their overlying atmosphere and to what extent can lakes modify their surrounding climate, and the uncertainties in these interactions'. A number of previous papers have addressed similar topics in the past (e.g. Dutra et al. 2010; Balsamo et al. 2012), but the strength of this current paper is the uncertainty estimation that it provides. Specifically, the authors introduce a third-party toolbox and the GLUE methodology to perform a sensitivity and uncertainty analysis of the different surface heat fluxes simulated by the Canadian Small Lake Model (CSLM), a one-dimensional integral lake model. The authors focus their study on a small lake in northern Canada, which is a good study site as small lakes are the most abundant at the global scale (see further my notes in the specific comments below). Within their sensitivity analysis, the authors find that the light attenuation coefficient, $K_d$, is the most important parameter controlling model performance and that variable $K_d$ provides the highest uncertainty in surface flux estimates.**

Thank you for the time you took understanding the paper.

**I don't particularly find this surprising, as others have found that water clarity can have a considerable influence on lake stratification and the turbulent heat fluxes (see Heiskanen et al. 2015) and can also considerably influence the diurnal cycles of heating and cooling in lakes (Woolway et al. 2016), but I do find this an important point to highlight and one that deserves some attention.**

Thank you for pointing out these two recent publications. We did make reference to other papers such as Fuentetaja et al. (1999) and Rinke et al. (2010) who highlight the importance of light attenuation in the functioning of the lake. We build upon these contributions by providing numerical estimates of the related uncertainties as well as qualifying and quantifying the importance of the light extinction coefficient.

We should definitely add more recent papers and we would like to thank the reviewer for pointing them out.

Added text:, p.2, l.34 *"The magnitude of these fluxes can be a function of several factors, such as lake area (Woolway et al., 2016), but the the clarity of the lake seems to be the dominant factor, especially for small lakes (Heiskanen et al., 2015; Woolway et al., 2016)."*

**While I think this paper will be of interest to those who focus on the integration of lakes within the climate system and for Numerical Weather Prediction, I strongly believe that the paper would be improved if there were more focus on the analysis and the results were put into context of the published literature.**

We agree that more emphasis should be put on the more recent literature and we would like to thank the reviewer for providing these relevant references.

Please see the added literature in response to the other re viewer's comments.

See in particular, p.3, l.5: "[...]*these measurements can be difficult and expensive and as such improved modelling approaches are necessary (e.g., McGloin et al., 2014b).*

**Often I found some of the most relevant literature being ignored and/or overlooked and some references, which were included in the text, seem inappropriate or irrelevant. One of my main criticisms is that a thorough literature review is needed to strengthen the introduction and discussion of the results. I provide some examples of relevant studies in this review, but there are many others which the authors should also look into. I strongly suggest a thorough review of the current literature prior to publication.**

Please refer to our previous comment.

**I find it surprising that the authors specify that Kd is the most important parameter controlling model performance, but do not include any detailed measurements of Kd. In particular, it is very likely that the lake has a different Kd to that estimated from the model sensitivity analysis.**

The reviewer is right. Please refer to our reply to the other anonymous reviewer where we stated that we did not make the measurements ourselves.

**Overall, I think there is some potential for this paper to be revised sufficiently to make it a valuable contribution to the scientific literature. However, addressing all of the points raised below are needed, in my opinion, prior to this paper being considered for publication in HESS.**

Thank you.

**Specific Comments Unfortunately the Downing et al. (2006) estimates of global lake size and abundance are no longer supported. Many studies have since shown that the Pareto distribution does not adequately describe the global distribution of lakes. For example, see Seekell and Pace (2011) and McDonald et al. (2012). A more detailed description of the global abundance and size distribution of lakes are provided by Verpoorter et al. (2014) and more recently by Cael and Seekell (2016). Granted that these recent studies to do not consider the smallest lakes of the world (for example, Verpoorter et al. only consider lakes larger than 0.002 km2), but still the authors should read up on these papers and include the relevant citations.**

p.1, l.17: Updated the estimate of total lake area and replaced Downing et al. (2006) with Verpoorter et al (2014) and Cael and Seekel(2016).

The text now reads: "*While lakes only cover around four percent of the Earth's land surface (Verpoorter et al., 2014; Cael and Seekell, 2016) [...]*"

**They also provide a more immediate feedback through mass and energy exchanges with the atmosphere' - you need some reference for this. As I'm sure you're aware, these fluxes are quite difficult to calculate (see Woolway et al. 2015a). Further information on these fluxes is needed, in my opinion. Additional information here will allow others who are not experts in the field to understand better the kind of interactions you are talking bout.**

Please refer to our previous comments agreeing that a revision of more recent literature is necessary, p.3, l.1:

"With different albedo, heat capacity and surface roughness than the surrounding land areas lakes also provide more immediate feedback through transfer of heat and moisture exchanges with the atmosphere (ex. MacKay et al. 2009; Xiao et al. 2013; McGloin et al. 2014). While some studies have performed direct measurements of latent and sensible turbulent heat fluxes from eddy covariance systems over lakes and reservoirs (e.g. Blanken et al. 2000; Vesala et al. 2006; Blanken et al. 2011; Nordbo et al. 2011 McGloin et al., 2014a) these measurements can be difficult and expensiveand as  such improved modelling approaches are necessary. (e.g.. McGloin et al. 2014)"

**Tanentzap et al. (2008) did not consider the influence of variations in thermocline depth on fluxes to the atmosphere, thus I don't think this reference is appropriate.**

Thank you for pointing out the error. –

"and ignore the internal thermal structure of lakes, which influences surface temperatures and thus fluxes to the atmosphere (Mackay 2012).

**'Rinke et al. (2010) illustrate the feedback between phytoplankton and thermal structure. . .' - There are other studies which you could also cite. For example, Mazumder et al. (1990) showed this over two decades ago. There are many other studies since then which I think the authors should read up on.**

We could add a few more references to this (e.g. Tilzer, 1983; Tilzer 1988; Mazmuder et al. 1990)

Tilzer M.M. (1983) The importance of fractional light absorption by photosynthetic pigments for phytoplankton productivity in Lake Constance. Limnology and Oceanography, 28, 833–846.

Tilzer M.M. (1988) Secchi disk – chlorophyll relationships in a lake with highly variable phytoplankton biomass. Hydrobiologia, 162, 163–171.

**It may also be worth mentioning that, on a regional scale, Samuelsson et al. (2010) found that the presence of lakes induces a warming on the European climate, and an observational study by Rouse et al. (2005) found that high-latitude lakes strongly enhance evapotranspiration when added to a landscape. A useful study, which I think the authors should cite, is Heiskanen et al. (2015). The authors should also look at the papers cited by Heiskanen et al. (2015) as these will be of direct relevant to this study. In addition, a paper by Rose et al. (2016) describes that water clarity can either amplify or suppress lake surface water temperatures, which in turn will influence their interaction with the atmosphere. Please read the Rose et al. (2016) paper and look at the references within.**

**A lake depth sensitivity analysis was undertaken by Balsamo et al. (2010) and might be worth mentioning also.**

Thank you for the updated references. Please refer to our previous comments regarding the need to update the paper.

 *"Accounting for these uncertainties could anchor the results of studies such as Samuelsson et al. (2010) and Rouse et al. (2005) who show how lakes impact regional climate."*

The other anonymous reviewer also stressed the relevance of Heiskanen and the reference was included.

**P2L26 - What is a small lake? How do you characterize a lake as small?**

There is a lot of variation on this in the literature. It varies from on the order of ~ 10 km2 (Verpoorter et al. 2014) to on the order of ~1km2 (McGloin et al. 2014).

This discussion was not taken in the article.

**P3L24 - Water clarity can have numerous other influences on lake temperatures. I think this section needs to be expanded. A few examples include its influence on the thermal structure of lakes (e.g. Persson and Jones 2008), its influence on the absorption of heat during the day and greater release in the evening leading to larger diurnal cycles (Woolway et al. 2015b) and influencing the likelihood of diurnal stratification as well as seasonal stratification. Also, studies have shown that surface waters have been browning over the last few decades (Roulet and Moore 2006). All of these points should be included and expanded.**

Please refer to our previous comments regarding these points.

Added text, p.4, l-2: "The underlying mechanisms are complex, even from a purely mechanistic perspective, water clarity affects lake hydrodynamics (Persson and Jones, 2008), even under a 24-hour period (Woolway et al., 2015). The effects also depend on lake morphometry (Woolway et al., 2016) and are further compounded by biogeochemical processes, such as browning waters (Roulet and Moore, 2006) and ecosystem function (Tilzer, 1983, 1988; Mazumder et al., 1990; Rinke et al., 2010)."

**Italics aren't needed for the description of all units.**

Agreed.

Fixed

**P12L30 - The authors state that Kd is not often measured and measuring Kd for every lake might be a practical impossibility. In my opinion, this is one of the largest uncertainties in the inclusion of lakes in NWP. For example, in ECMWF's IFS Kd is assumed equal to 3 for all lakes, which could result in numerous biases in the turbulent heat fluxes. While I somewhat agree with the author's statements here, it may also be worth mentioning that satellites can estimate Kd, so there are possibilities in improving lake surface water temperature simulations. For more information, see Torbick et al. (2013) for information on how satellites can potentially be used to estimate secchi depth, which can be used as an indicator of Kd.**

Thank you for the references.

Added text, p.14, l.8: *"Remote sensing might provide a solution to lack of in situ measurements, and is in fact used to provide estimates of the Secchi depth (Torbick et al., 2013), a proxy for the light extinction coefficient."*

**P13L3 - 'this kind of monitoring has never been performed' - This isn't true. Lake monitoring stations now often have light sensors above and below the water surface and are thus used to determine water clarity and Secchi depth observations are traditionally recorded. I suggest the authors look through the literature to find examples of where they've been used. I'm almost certain that this information has not been used in NWP or climate modeling, but I hope in the future meteorologists and limnologists will work closer to address this and similar issues. A literature search on this topic is also needed in my opinion.**

Our statement was untrue. It should however be fair to say that such measurements are not commonplace, especially when it comes to continuous measurements.

Removed the statement and added instead, p.13, l.12: *"Such measurements have proved useful in evaluating turbulent transfers over large lakes (Deacu et al., 2012).*

**P13L9 - I don't think this can be a main conclusion as unfortunately it is not unknown. For example, see Heiskanen et al. (2015).**

It can be instead expressed as confirming and building upon Heiskanen's results.

Did not set up as main conclusion and deleted the text according to the following item in this review.

**There doesn't appear to be much discussion in this paper. I would recommend restructuring the paper to include separate 'Results' and 'Discussion' sections and perhaps reduce the conclusion to one or two paragraphs. This, in my opinion, would make the paper easier to digest.**

The conclusions should definitely be rewritten in light of the more recent literature.

Deleted from conclusions: *"Therefore the main conclusion of this paper was that measurements of K d might be necessary if the linkage between lakes and the atmosphere is to be improved. Furthermore the extinction coefficient is known to be temporally variable due to, amongst other factors, the vegetation cycle in lakes as well as sediment transport [Tanentzap et al., 2008; Rinkeet al., 2010], emphasizing the need for continuous monitoring.*

*The two main caveats against a generalizing conclusion regarding heat transfer from small lakes to the atmosphere are related to: a)the geomorphological characteristics of thelake and b) the limitations of SA methods. On the geomorphological side, the studied lake was shallow, probably minimizing importance of turbulent transfer in the modeling of heat fluxes and therefore studies on lakes with different characteristics should be the subject of future work in order to generalize the results found here. Furthermore, it should be said that it was assumed that quantitiessuch as lake area and average depth were accurately measured and might have an impact on simulation results, were those assumptions proven wrong."*

Added to conclusions, p13. l.27: *"This is perhaps not surprising given the recognized importance of the light extinction coefficient in modulating heat transfers (Heiskanen et al., 2015; Woolway et al., 2016; Rose et al., 2016). Given the complex and intertwined processes that can affect light penetration, such as browning waters (Roulet and Moore, 2006) and ecosystem function (Tilzer, 1983, 1988; Mazumder et al., 1990; Rinke et al., 2010), a single measurement of its value might*

*prove insufficient. It might be necessary to rely on continous measurement in order to improve modeling, either through adherence to a parsimony principle (its value need not be modelled if actually measured) or stemming from the need of evaluating the complex processes influencing it value.*"

**I don't find many of the figures presented in the paper very informative. They seem to all show similar results. Much of this information could be shown in 1 or 2 figures, in my opinion.**

It is one of the points of the paper that the results of the different SA methods are similar. This reinforces the importance of the light attenuation coefficient.

**Figure 1 needs more information. For example, can the authors add a smaller inset map to show where the lake is? Also, the figure would need a 'scale ruler' so that the reader can easily interpret the size of the lake.**

Please refer to our reply to a similar comment from the other anonymous reviewer.

**At first glance, I don't particularly understand Fig. 8. It isn't clear what the grey regions represent as one would expect the grey area to be an envelope that surrounds the main (blue) line?**

We could include a better description of the GLUE methodology and what the uncertainty bounds represent. An envelope around the main blue line should not be expected if the model fails to adequately reproduce the variable.

Here is the description in the paper, p.10, l.22:

*GLUE starts with a random sampling of the parameter space and subsequent computation of the simulation-performance for each random parameter. The random runs are then classified into behavioral (well-performing) or non-behavioral according to either subjective (e.g. threshold value) or objective criteria (limits-of-acceptability; Coxon et al. (2014). The behavioral simulations are then weighted according to the performance and uncertainty bounds extracted from weighted simulations: e.g. at each time step the 0.05 and 0.95 percentiles of the likelihood weighted simulations can be extracted and considered to be the 95% confidence interval.*

**Fig. 9 - Isn't irradiance a term often used to describe solar irradiance and not the turbulent fluxes? Also, why isn't there an x-label on the bottom panels?**

The x-label was placed on top instead. It is the parameter name.

The reviewer is right, although the units are the same, irradiance is often thought to refer solely to solar irradiance. The caption could read "heat-flux"

Fixed.

**Fig. 10 - I'm not sure how to interpret this figure. Can you please provide a better description of what we're seeing? I think a more detailed discussion of this figure should be given in the text.**

Reviewer one also suggested to make better use of this figures. It illustrates the trade-offs made by the model when simulating latent and sensible heat.

Please see reply to first reviewer,

**Table 1 - The square brackets appear the wrong way round in the fourth column.**

The brackets were consciously placed as such to show that the set does not include its bounds.

**References:**

Balsamo G, et al (2010), Deriving an effective lake depth from satellite lake surface temperature data: a feasibility study with MODIS data. Boreal Environment Research 15:178-190.

Balsamo G, Salgado R, Dutra E, Boussetta S, Stockdale T, Potes M (2012), On the contribution of lakes in predicting near-surface temperature in a global weather forecasting model. Tellus A 64, 15829.

Cael BB, Seekell DA (2016), The size-distribution of Earth's lakes. Sci Rep 6, 29633.

Dutra E, Stepanenko VM, Balsamo G, Viterbo P, Miranda PM, et al. (2010), An offline study of the impact of lakes on the performance of the ECMWF surface scheme. Boreal Env. Res. 15:100–112.

Heiskanen JJ, et al. (2015), Effects of water clarity on lake stratification and lakeatmosphere heat exchange. J Geophys Res Atmos 120:7412-7428

Mazumder A, Taylor WD, McQueen DJ, Lean DR (1990), Effects of fish and plankton and lake temperature and mixing depth. Science 247:312–315

McDonald CP, et al. (2012), The regional abundance and size distribution of lakes and reservoirs in the United States and implications for estimates of global lake extent. Limnol. Oceanogr. 57:597-606.

Persson I, Jones ID (2008) The effect of water colour on lake hydrodynamics: a modeling study. Freshwater Biol 53:2345-2355

Rose KC, Winslow LA, Read JS, Hansen GJA (2016) Climate-induced warming of lakes can be either amplified or suppressed by trends in water clarity. Limnol Oceanogr Lett 1:44-53

Roulet N, Moore TR (2006) Environmental chemistry: Browning the waters. Nature 444:283–284.

Seekell DA, Pace ML (2011), Does the pareto distribution adequately describe the size-distribution of lakes? Limnol. Oceanogr. 56(1):350-356.

Torbick N, Hession S, Hagen S, Wiangwang N, Becker B, Qi J (2013) Mapping inland lake water quality across the Lower Peninsula of Michigan using Landsat TM imagery. Int J Remote Sens, 34:7607–7624.

Verpoorter C, Kutser T, Seekell DA, Tranvik LJ (2014), A global inventory of lakes based on high-resolution satellite imagery. Geophys Res Lett 41:6396-6402.

**Woolway, R.I, Jones, I.D., Hamilton, D.P. et al. (2015a). Automated calculation of surface energy fluxes with high-frequency lake buoy data. Environmental Modelling & Software 70, 191-198.**

**Woolway, R.I., Jones, I.D., Feuchtmayr, H. et al. (2015b). A comparison of the diel variability in epilimnetic temperature for five lakes in the English Lake District. Inland Waters 5(2), 139-154.**

**Woolway, R.I., Jones, I.D., Maberly, S.C. et al. (2016). Diel surface temperature range scales with lake size. PLoS One 11(3): e0152466. doi: 10.1371/journal.pone.0152466**

**Editor's comments**

It was an interesting discussion, enthusiastic (and at places quite critical) comments from referees and replies from the authors. I am inviting the authors to update the manuscript accordingly, and to submit it together with rebuttal (based on already published replies). I would like to add one thing, for possible consideration: Figure 9 (results of GLUE) shows that MAE (and hence model output) does not depend on values of Cf and Cs (and it is also stated on p 12). How physical is this? How does this Figure relate to Figures 6-7 where one can see that MAE or NSE are indeed sensitive to Cs and Cf?

Thank you for your comments. We agree on the positive nature of the feedback provided by the reviewers.

Cf and Cs have to do with turbulent transport and dissipation respectively. From a physical point of view, this might reflect the fact that the lake is rather shallow and that a thermocline might not develop, thus modulating the impact of these parameters.

On the other hand this might also be due to numerical issues and the inherent limitations and assumptions of different sensitivity analysis methods. This underlines the need for testing different methods and not only relying on one. A particular weakness of FAST is its reliance on periodic sampling, making it difficult to cover the parameter space in its entirety. Pinpointing the exact reason for the difference would require a more in-depth study and might in the end be down to numerics.

---

## Author Response (AR2)

**Reply to reviewers**

We would like to thank the editor Dimitri Solomatine and the two anonymous reviewers for their constructive comments, which have helped make this a better paper. Here, we reply to the latest round of comments. The reviewer's comment are in bold font.

**Reviewer 1:**

**P3 l25 consider "…depth interval."**

Done.

**P4 l2 -the underlying mechanisms controlling what? It was not clear to me whether this sentence referred to heat fluxes or k_d**

We meant the mechanisms controlling lake temperature, which in turn modulate heat fluxes. The sentence was changed to read:

"The mechanisms underlying the thermal structure of lakes are complex. From a purely mechanistic perspective water clarity affects lake hydrodynamics[...]. The thermal structure further depends on lake morphometry […] and is compounded by biogeochemical processes [...]"

**P4 l5 –please list the relevant aspects of ecosystem function**

We already list browning waters, and ecosystem functions. Details of this aspect are given in the provided references. For instance, plankton concentrations will affect light attenuation. Browning waters directly affect water color and hence light absorption characteristics.

**P5 l22 "… using a krypton …"**

Done.

**P6 l6 "… evaluation data and computational…"**

Done, but using Oxford comma.

**P6 l6 "… 2006): different …"**

Done.

**P6 l12 – there is an incomplete sentence here, maybe "… state of the art." ?**

Yes, the sentence is incomplet. Done.

**P6 l24 "…criteria: the …"**

Done.

**P6 l29-30 – does GLUE always require uniform random sampling or this just a convention?**

The rationale is that the distribution used for the sampling should reflect prior knowledge about the distribution of the parameter. Using an uniform distribution equates to assuming no prior knowledge about the parameter values beyond giving it a plausible range. GLUE can be used with other sampling schemes, such as membership functions.

**P7 l13-14 – this sentence could be deleted as it effectively repeats p7l1-2**

Done.

**P7 l19-22 – please consider splitting this into two sentences.**

Done.

**P8 l16 "…framework: they …"**

Done.

**P8 l21 "… as was …"**
Done.
**P8 l21-22 – please expand on this sentence. How is robustness dependent on predictive capability?**
A detailed explanation is given in the provided reference. Citing from it:

" The advantage of this method is that it is straightforward and simple to apply. However, it is not applicable when the relationship between the factors and model output is non-linear or non-monotonic and when there is a high level of interactions among factors. As such, regression analysis often performs poorly. For non-linear models, the rank transformation can be helpful since the rank transformation can cope with non-linear models and mitigate the detrimental effect of long tailed output distribution (Saltelli and Sobol', 1995). The rank transformation also has two drawbacks: firstly it fails with non-monotonic models, and secondly its main effect is a forced linearization of the system by an artificial increase in the relative weight of the first order terms (Saltelli and Sobol', 1995) so the result cannot be transformed back to the original model. In this study the rank transformation is also applied for reference purposes."

**P9 l2 – extra line break after "…are:"**
Done.
**P9 l30 "… indices (Sobol…"**
Done.
**P10 l10 –consider rephrasing as "… summands are orthogonal …" or "Assuming orthogonality of all summands …"**
Done.
**P10 l11 – "allows" works but "makes it possible" works better**
Done.
**P10 l24 – missing close bracket "… Coxon et al. (2014)).**
Done.
**P11 l5 "… available for …"**
Done.
**P12 l14 – I've read Duan et al. (2006) and took it for granted that response surfaces are complicated, but my experience with high dimensional models suggests that response surfaces are often quite simple. As this submission is a relatively short paper, I would be interested in reading the authors' thoughts on Duan's assertion in light of their results, which seem to suggest a relatively simple response surface for a complex model**
This is difficult to answer with finality.

For some models, exploring the response surface in its entirety is a practical impossibility. If only a few parameters are considered in the analysis then the response surface might become simple, but perhaps artificially so. When calibrating a large number of parameters, we have often found the calibration never to converge, which can be attributed to either a complex response surface and/or numerical issues.

It could be argued that the response surface for this lake model appears simple because it is dominated by the sensitivity to the light extinction coefficient, which shadows all other parameters. It might not be the case if other parameters proved themselves to be more sensitive.

**P12 l19 – case agreement, either "… coefficients exert …" or "…coefficient exerts …"**

Done. Coefficients.

**P13 l8 "… indicative of effective …"**

Done.

**Reviewer 2:**

**I think that the authors have done a good job in addressing my earlier comments and the paper is now suitable for publication. I very much enjoyed reading the latest version and found the structure and flow of the manuscript much better. I'm confident that the limnological community will appreciate this research and the work will be well-cited.**

Thank you for the kind comment.

**Although I could not find any major issues with the paper, I did find areas where minor corrections are needed:**

**For example, Page 3 Line 14 there should be a space between 1.12 and km2**

Done.

**Page 2 Line 21 - Please don't start a new sentence with citations. It reads much better if you would say something along the lines of 'previous studies have illustrated the feedback between phytoplankton... (REFS)'**

Done.

"Previous studies illustrate [...]"

**Page 2 Line 30 - There is a paper recently published in GRL which demonstrates how the stability of the atmospheric boundary layer varies among lakes (latitude and lake size are important predictors of over-lake atmospheric stability by Wollway et al doi:10.1002/2017GL073941), which would be a very useful citation here, as it demonstrates how lakes can modify the boundary layer, thus strengthening your argument.**

Added text:

"[...] and the latitude of the lake (Woolway et al., 2017)"

**Page 7 Line 17 - 'as done by (Deacu et al., 2012) for large lakes'. Note that there shouldn't be any brackets around the entire citation here, so you will need to change the command in your latex file. Please check the entire manuscript for inconsistencies such as this.**

Fixed.

**Page 3 Line 8 - 'which can in turned...' -> replace with 'in turn'**

Fixed.

**Page 3 Line 30 - include Kd in the brackets along with m^-1**

Fixed.

**Page 10 Line 24 - (limits-of-acceptability... -> need to close the brackets here.**

Fixed.

**Page 11 Line 5 - 'that were available f for' -> remove extra 'f'?**

Fixed.

**Please take extra care when submitting the final version for publications as there may be other errors (such as above) still present in the paper.**

**The authors use different citation formats throughout the text. For example, sometimes 'et al.' is used and other times 'et al.,' is used. This is me being very picky but please try to be consistent throughout.**

This is consistent withing the style file provided by HESS. It uses a comma when separating the year of publication.

**The concluding paragraph is one of the most important parts of the paper. Please consider re-writing into a concise format (the short sentence on its own at the end should be merged with the previous paragraph, if possible).**

**Sometimes the authors use 'and' in citations and other times use '&' - please be consistent.**

Done. Editor names left as is: e.g. "Taylor & Francis"

**Page 8 Line 15 - need to close bracket... '(SRC: Galton (1886) share'**

Fixed.

**Table 1 - The brackets are pointing the wrong way in the column 'Range'. Please correct. Also, please prove a better description in the table caption (currently not sufficient).**

They point in the right direction as they indicate a non-inclusive interval.

**Figure 1 is currently very hard to see in detail. For example, the inset and text is too small. Please increase the figure size in the final version.**

I believe the size of the figure is larger in the submitted file but as been scaled to fit in the pdf.

**Many of the Figure captions need to be improved. Specifically, the reader should not have to look through the manuscript to understand the figures. In my opinion the reader should be able to see the figure, read the caption, and understand what is presented. Please include a more detailed description in each caption. This will help the reader and thus make the paper easier to digest and follow, especially for those who will glance the paper.**

Changed captions for most figures.

**Consider colorblind friendly colors for Fig. 6 and 7.**

Replaced the jet colormap with the parula colormap. Hopefully this helps.

**Table 2 - need space before bracket in references. Also, please ensure reference style is consistent with the remainder of the paper and include a fuller description in the caption.**

Fixed.

**Figure 9 - Include units on y and x-axis (not just in the caption). This is only necessary for the outer panels.**

Fixed

**Fig 10 (and throughout text) I would suggest replacing Mean Absolute Error with Mean Absolute Difference. Also, W/m2 needs to be in a consistent format to elsewhere in the text, currently Wm-2 elsewhere.**

Changed to W.m-2. Mean Absolute Error and Difference are interchangeable.

**Table 3 - include comma to separate the 1000s**

We feel the order of magnitude is small enough as to not require thousands separation.